# Tracking Most Significant Shifts in Infinite-Armed Bandits

**Joe Suk** [1]   **Jung-hun Kim** [2]

## Abstract

We study an infinite-armed bandit problem where actions' mean rewards are initially sampled from a *reservoir distribution*. Most prior works in this setting focused on stationary rewards (Berry et al., 1997; Wang et al., 2008; Bonald and Proutiere, 2013; Carpentier and Valko, 2015) with the more challenging adversarial/non-stationary variant only recently studied in the context of rotting/decreasing rewards (Kim et al., 2022; 2024). Furthermore, optimal regret upper bounds were only achieved using parameter knowledge of non-stationarity and only known for certain regimes of regularity of the reservoir. This work shows the first parameter-free optimal regret bounds while also relaxing these distributional assumptions. We also study a natural notion of *significant shift* for this problem inspired by recent developments in finite-armed MAB (Suk and Kpotufe, 2022). We show that tighter regret bounds in terms of significant shifts can be adaptively attained. Our enhanced rates only depend on the rotting non-stationarity and thus exhibit an interesting phenomenon for this problem where rising non-stationarity does not factor into the difficulty of non-stationarity.

## 1. Introduction

We study the multi-armed bandit (MAB) problem, where an agent sequentially plays arms from a set $\mathcal{A}$, based on partial and random feedback for previously played arms called rewards. The agent's goal is to maximize earned rewards.

Much of the classical literature (see Bubeck and Cesa-Bianchi, 2012; Slivkins, 2019; Lattimore and Szepesvári, 2020, for surveys) focuses on finite armed bandits where $\mathcal{A} = [K]$ for some fixed $K \in \mathbb{N}$. The theory here then typically assumes a large time horizon of play $T$ relative to

$K$. However, in practice, the number of arms can be prohibitively large as is the case in recommendation engines or adaptive drug design motivating the so-called *many-armed*, or *infinite-armed*, model.

At the same time, another practical reality is that of changing reward distributions or *non-stationarity*. While there has been a surge of works here (Kocsis and Szepesvári, 2006; Yu and Mannor, 2009; Garivier and Moulines, 2011; Mellor and Shapiro, 2013; Liu et al., 2018; Auer et al., 2019; Chen et al., 2019; Cao et al., 2019; Manegueu et al., 2021; Wei and Luo, 2021; Suk and Kpotufe, 2022; Jia et al., 2023; Abbasi-Yadkori et al., 2023; Suk, 2024), most works here again focus on the finite-armed problem.

This work studies infinite-armed bandits where mean rewards of actions are initially drawn from a *reservoir distribution* and evolve over time under rested non-stationarity. While much of the existing literature on this topic focuses on stationary rewards (Berry et al., 1997; Wang et al., 2008; Bonald and Proutiere, 2013; Carpentier and Valko, 2015), the more challenging adversarial or non-stationary scenario has only recently been explored in the context of rotting (i.e., decreasing) rewards (Kim et al., 2022; 2024). Kim et al. (2024)'s state-of-the-art algorithm for rotting bandits relies on prior knowledge of non-stationarity parameters and further regularity assumptions on the reservoir distribution to attain optimal regret bounds. However, these assumptions can be impractical in real-world applications.

This work studies a broader non-stationary model where rewards are decided by an adaptive adversary and aims to derive regret bounds without requiring algorithmic knowledge of non-stationarity. Additionally, we go beyond the task of attaining optimal regret bounds as posed by Kim et al. (2024), and show enhanced regret bounds which can be tighter (i.e., possibly near-stationary rates) despite large non-stationarity, thus more properly capturing the theoretical limits of learnability in changing environments. Such insights are inspired by similar developments in the finite-armed analogue (Suk and Kpotufe, 2022), where substantial changes in best arm, called *significant shifts*, can be detected leading to more conservative procedures which don't overestimate the severity of non-stationarity.

---

[1]Columbia University [2]CREST, ENSAE Paris. Correspondence to: Joe Suk <joe.suk@columbia.edu>.

*Proceedings of the 42nd International Conference on Machine Learning*, Vancouver, Canada. PMLR 267, 2025. Copyright 2025 by the author(s).

## 1.1. More on Related Works

The most relevant works are Kim et al. (2022; 2024). The first of these works studies a simpler model where the reservoir distribution is uniform on $[0, 1]$ and there's a fixed upper bound $\rho$ on the magnitude of round-to-round non-stationarity. Kim et al. (2022) show the minimax regret rate is $\rho^{1/3} \cdot T + \sqrt{T}$ and derive a matching upper bound, up to log terms, but using algorithmic knowledge of $\rho$ as well as a suboptimal regret upper bound without knowledge of $\rho$.

Kim et al. (2024) study a general setting where the reservoir distribution for initial mean reward $\mu_0(a)$ of arm $a$ satisfies $\mathbb{P}(\mu_0(a) > 1 - x) = \Theta(x^\beta)$ for all $x \in (0, 1)$. They study a more general setting of 1-sub-Gaussian random rewards vs. our setting of $[0, 1]$-bounded rewards. They show a regret lower bound of order $\min\{V^{\frac{1}{\beta+2}} \cdot T^{\frac{\beta+1}{\beta+2}}, L^{\frac{1}{\beta+1}} \cdot T^{\frac{\beta}{\beta+1}}\}$ in terms of number of changes $L$ in rewards or total variation $V$ (quantifying magnitude of changes over time).

With knowledge of $L$ and $V$, they show a matching regret upper bound for $\beta \geq 1$ and a worse $\min\{V^{1/3}T^{2/3}, \sqrt{LT}\}$ bound for $\beta < 1$. We suspect this latter bound for $\beta < 1$ is in fact tight due to earlier works (Carpentier and Valko, 2015, Theorem 1) showing $T^{-1/2}$ lower bounds on the final-round regret. Interestingly, in our $[0, 1]$-bounded rewards setting, this phenomenon does not occur and the minimax regret behaves the same for $\beta < 1$ or $\beta \geq 1$. Finally, Kim et al. (2024) also have suboptimal regret upper bounds without knowledge of $L$ or $V$.

We also note that non-stationarity is well-studied in more structured many or infinite-armed bandit settings such as (generalized) linear (Cheung et al., 2019; Russac et al., 2020; Zhao et al., 2020; Kim and Tewari, 2020; Wei and Luo, 2021), kernel (Hong et al., 2023; Iwazaki and Takeno, 2024), or convex bandits (Wang, 2022). To contrast, our infinite-armed setting does not assume any metric structure on the arm space and so these works are not easily to comparable to this paper. We also note there are no known results on more nuanced measures of non-stationarity, like the significant shifts, even for such structured settings.

## 1.2. Contributions

- We show the first optimal and adaptive (a.k.a. parameter-free) regret upper bounds for non-stationary infinite-armed bandits (Theorem 5). In fact, our bounds are expressed in terms of tighter and more optimistic measures of non-stationarity (Subsection 2.2 and Theorem 3) new to this work. This resolves open questions of Kim et al. (2022; 2024).

- We note our procedures are substantially different from Kim et al. (2022; 2024) who rely on explore-then-exploit strategies whereas we revisit the subsampling approach of Bayati et al. (2020) which partially reduces

the problem to analyzing finite-armed bandits.

- Along the way, we develop the first high-probability regret bounds for the infinite-armed setting. Our regret upper bound in Theorem 5 also relaxes distributional assumptions on the reservoir distribution, not requiring an upper bound on the masses of randomly sampled rewards. Both such generalizations were unknown in prior works even in the stationary setting.

- To our knowledge, our work is the first to develop adaptive dynamic regret bounds of the style $V^{1/3}T^{2/3} \wedge \sqrt{LT}$ with bandit feedback and adaptively adversarial changes. Notably, such results are yet unknown in the finite-armed setting. Note our setting is not directly comparable to finite-armed bandits as we assume a known optimal reward value.

- We validate our findings via experiments on synthetic data, showing our procedures perform better than the previous art for rotting infinite-armed bandits.

## 2. Setup

### 2.1. Non-Stationary Infinite-Armed Bandits

We consider a multi-armed bandit with infinite armset $\mathcal{A}$. At each round $t$, the agent plays an arm $a_t$, choosing either to newly sample $a_t$ from $\mathcal{A}$ or to play an already sampled arm among the previously chosen arms $\{a_1, \ldots, a_{t-1}\}$.

When the agent samples arm $a_t \in \mathcal{A}$ at round $t$, it observes a random reward $Y_t(a_t) \in [0, 1]$ with mean $\mu_t(a_t) \in [0, 1]$ whose value is randomly drawn from a *mean reservoir distribution*. The reward of this chosen arm in the subsequent round is then decided according to an adaptive adversary with access to prior decisions $\{a_s\}_{s \leq t}$ and observations $\{Y_t(a_s)\}_{s \leq t}$. As in prior works (Berry et al., 1997; Wang et al., 2008; Carpentier and Valko, 2015; Bayati et al., 2020; Kim et al., 2024), we assume the reservoir distribution is $\beta$-regular, parametrized by a shape parameter $\beta > 0$.

**Assumption 1** ($\beta$-Regular Reservoir Distribution). *We assume a $\beta$-**regular** reservoir distribution for some $\beta > 0$: there exists constants $\kappa_1, \kappa_2 > 0$ such that for all $x \in [0, 1]$:*

$$\kappa_1 \cdot x^\beta \leq \mathbb{P}(\mu_0(a) > 1 - x) \leq \kappa_2 \cdot x^\beta.$$

**Remark 1.** *Prior works on non-stationary bandits (e.g. Besbes et al., 2019) typically allow for unplayed arms' rewards to change each round, which is equivalent to our setting with a different accounting of changes, as changes in yet unplayed arms do not affect performance.*

Let $\delta_t(a) := 1 - \mu_t(a)$ be the gap of arm $a$ at round $t$. Then, the *cumulative regret* is $\mathbf{R}_T := \sum_{t=1}^{T} \delta_t(a_t)$. Let $\delta_t(a, a') := \mu_t(a) - \mu_t(a')$ be the *relative regret* of arm $a'$ to $a$ at round $t$.

## 2.2. Non-Stationarity Measures

Let $V := \sum_{t=2}^{T} |\mu_{t-1}(a_{t-1}) - \mu_t(a_{t-1})|$ denote the *realized total variation* which measures the total variation of changes in mean rewards through the sequence of played arms. In Section 5, we also consider the *total realized rotting variation* $V_R := \sum_{t=2}^{T} (\mu_{t-1}(a_{t-1}) - \mu_t(a_{t-1}))_+$ which sums the magnitudes of rotting in rewards over time.

Let $L := \sum_{t=2}^{T} \mathbf{1}\{\mu_t(a_{t-1}) \neq \mu_{t-1}(a_{t-1})\}$ denote the *realized count of changes*, and $L_R := \sum_{t=2}^{T} \mathbf{1}\{\mu_t(a_{t-1}) < \mu_{t-1}(a_{t-1})\}$ the *realized number of rotting changes*

To contrast, the prior works Kim et al. (2022; 2024) consider a priori upper bounds on $V, L$ (i.e., the adversary is constrained to incur non-stationarity at most size $V$ or count $L$), and so only show expected regret bounds in terms of such bounds. Our work establishes stronger high-probability regret bounds in terms of our tighter realized values of $V, L$.

**Remark 2.** *We note that our measures of non-stationarity $V, V_R, L, L_R$ depend on the agent's decisions and so may not be directly comparable for different algorithms and/or adversaries. For an oblivious adversary, these quantities can be upper bounded by worst-case analogues which do not depend on the agent's decisions. Even in this more limited setting, optimal and adaptive regret upper bounds were previously unknown.*

# 3. Regret Lower Bounds

Kim et al. (2024, Theorem 4.1 and 4.2) show regret lower bounds of order $(L_R+1)^{\frac{1}{\beta+1}} \cdot T^{\frac{\beta}{\beta+1}} \wedge (V_R^{\frac{1}{\beta+2}} \cdot T^{\frac{\beta+1}{\beta+2}} + T^{\frac{\beta}{\beta+1}})$ for the rotting infinite-armed bandit problem, which is a subcase of our non-stationary setup. Our regret upper bound in Theorem 5 matches this lower bound up to $\log$ factors, without algorithmic knowledge of $L_R, V_R$.

# 4. A Blackbox for Optimally Tracking Unknown Non-Stationarity

## 4.1. Intuition for Subsampling

A key idea, used for the stationary problem in Wang et al. (2008); Bayati et al. (2020), is that of *subsampling* a fixed set of arms from the reservoir. The main algorithmic design principle is to run a finite-armed MAB algorithm over this subsample. The choice of subsample size is key here and exhibits its own exploration-exploitation tradeoff, appearing through a natural regret decomposition with respect to the subsample, which we'll denote by $\mathcal{A}_0 \subseteq \mathcal{A}$:

$$\mathbf{R}_T = \underbrace{\sum_{t=1}^{T} \min_{a \in \mathcal{A}_0} \delta_t(a)}_{\text{Regret of best subsampled arm}} + \underbrace{\sum_{t=1}^{T} \max_{a \in \mathcal{A}_0} \delta_t(a, a_t)}_{\text{Regret to best subsampled arm}} .$$

(1)

Suppose there are $K := |\mathcal{A}_0|$ subsampled arms. One can show (e.g., Theorem 11) a size $K$ subsample of a $\beta$-regular reservoir contains, with high probability, an arm with gap $O(K^{-1/\beta})$. Thus, the first sum in (1) is at most $T \cdot K^{-1/\beta}$.

Then, plugging in the classical gap-dependent regret bounds for finite-armed MAB, the second sum in (1) is $\tilde{O}\left(\sum_{i=2}^{K} \Delta_{(i)}^{-1}\right)$ where $\Delta_{(i)}$ is the (random) $i$-th smallest gap to the best subsampled arm. Then, it is further shown (Bayati et al., 2020, Section A.2) that this gap-dependent quantity scales like $\tilde{O}(K)$ in expectation, by carefully integrating $\Delta_{(i)}^{-1}$ over the randomness of the reservoir.

Then, choosing $K$ to balance the bounds $T \cdot K^{-1/\beta}$ and $K$ on (1) yields an optimal choice of $K \propto T^{\frac{\beta}{\beta+1}}$ giving a regret bound of $T^{\frac{\beta}{\beta+1}}$ which is in fact minimax.

**Key Challenges:** Extending this strategy to the non-stationary problem, it's natural to ask if we can follow an analogous strategy by reducing to $K$-armed non-stationary bandits. However, this poses fundamental difficulties:

(a) As our goal in the non-stationary problem is to achieve *adaptive* regret bounds, without parameter knowledge, a naive approach is to reduce to adaptive $K$-armed non-stationary MAB guarantees (Auer et al., 2019; Wei and Luo, 2021; Suk and Kpotufe, 2022; Abbasi-Yadkori et al., 2023). However, these guarantees only hold for an oblivious adversary, and so are inapplicable to our problem. Furthermore, these algorithms only give worst-case rates of the form $\sqrt{LKT}$ in terms of $L$ changes. In fact, it's known in this literature that no algorithm can adaptively secure tighter gap-dependent rates over unknown changepoints (Garivier and Moulines, 2011, Theorem 13). As the gap-dependent regret bound is crucial to achieving optimally balancing exploration and exploitation in our subsampling strategy, we see this approach can only hope to achieve suboptimal rates.

(b) Secondly, we observe that upon experiencing changes, one may have to *re-sample* arms from the reservoir distribution as the regret of the best subsampled arm $\min_{a \in \mathcal{A}_0} \delta_t(a)$ can itself become large over time. Thus, we require a more refined subsampling strategy which works in tandem with non-stationarity detection.

## 4.2. Our New Approach: Regret Tracking

We handle both of the above issues with the new idea of tracking the empirical regret $\hat{\delta}_t(a_t) := 1 - Y_t(a_t)$ as a proxy for tracking non-stationarity. The key observation is that the empirical cumulative regret of played actions up to round $t$, $\sum_{s=1}^{t} \hat{\delta}_s(a_s)$, concentrates around $\sum_{s=1}^{t} \delta_s(a_s)$ at fast logarithmic rates by Freedman's inequality and a self-bounding argument for $[0, 1]$-valued random variables.

This means, so long as $\sum_{s=1}^{t} 1 - Y_s(a_s) \lesssim t^{\frac{\beta}{\beta+1}}$, our regret will be safe up to the minimax stationary regret rate. On the other hand, if $\sum_{s=1}^{t} 1 - Y_s(a_s) \gg t^{\frac{\beta}{\beta+1}}$, then the agent must be experiencing large regret which means some non-stationarity has occurred if the agent otherwise plays optimally for stationary environments.

Thus, at a high level, our procedure (Algorithm 1) restarts the subsampling strategy outlined in Subsection 4.1 upon detecting large empirical regret.

Setting up relevant terminology, an *episode* is the set of rounds between consecutive restarts and, within each episode, we further employ doubling epochs, termed *blocks*, to account for unknown changepoints and durations of play.

Within each block, we run the subsampling strategy for a fixed time horizon as a blackbox. The blackbox takes as input a finite-armed MAB base algorithm, parametrized by Base-Alg$(t, \mathcal{A}_0)$ for inputs horizon $t$ and subsampled set of arms $\mathcal{A}_0 \subset \mathcal{A}$.

Our only requirement of the base algorithm is that it attains a gap-dependent regret bound in so-called *mildly corrupt* environments, defined below. It's straightforward to show this is satisfied by classical stochastic MAB algorithms such as UCB (Lai and Robbins, 1985) (proof in Section C).

**Definition 1.** *We say a finite-armed non-stationary bandit environment $\{\mu_t(a)\}_{t \in [T], a \in \mathcal{A}_0}$ over horizon $T$ with armset $\mathcal{A}_0$ is $\alpha$-**mildly corrupt** for $\alpha > 0$ if there exists a **reference reward profile** $\{\mu(a)\}_{a \in \mathcal{A}_0}$ such that*

$$\forall t \in [T], a \in \mathcal{A}_0 : |\mu_t(a) - \mu(a)| \leq \alpha.$$

Next, in stating the requirement of our base algorithm, we use $\delta_t(a, a') := \mu_t(a) - \mu_t(a')$ to denote the gap of arm $a'$ to $a$ in the context of a finite-armed bandit $\{\mu_t(a)\}_{t \in [T], a \in \mathcal{A}_0}$.

**Assumption 2.** *Let $\{\mu_t(a)\}_{t \in [T], a \in \mathcal{A}_0}$ be an $\alpha$-mildly corrupt $T$-round finite-armed bandit with reference reward profile $\{\mu(a)\}_{a \in \mathcal{A}_0}$. Let $\Delta_{(2)} \leq \Delta_{(3)} \leq \cdots \Delta_{(|\mathcal{A}_0|)}$ be the ordered gaps induced by the reference reward profile. Then, Base-Alg$(T, \mathcal{A}_0)$ attains, with probability at least $1 - 1/T$, for all $t \in [T]$, a $t$-round static regret bound of (for $C_0$ free of $t, T, \mathcal{A}_0$):*

$$\max_{a \in \mathcal{A}_0} \sum_{s=1}^{t} \delta_s(a, a_s) \leq 6t\alpha + C_0 \sum_{i=2}^{|\mathcal{A}_0|} \frac{\log(T)}{\Delta_{(i)}} \mathbf{1}\{\Delta_{(i)} \geq 4\alpha\},$$

**Remark 3.** *Our Assumption 2 is at firt glance similar to Assumption 1 of Wei and Luo (2021), but it in fact stronger in requiring gap-dependent bounds in environments with small variation as opposed to their requirement of $O(\sqrt{T})$ regret in such environments (Wang, 2022, Lemma 3).*

**Remark 4.** *Bandit algorithms attaining state-of-the-art regret bounds in **stochastic regimes with adversarial corruption** (Lykouris et al., 2018; Gupta et al., 2019; Zimmert*

*and Seldin, 2019; Ito, 2021; Ito and Takemura, 2023; Dann et al., 2023) satisfy Assumption 2.*

---

**Algorithm 1:** Blackbox Non-Stationary Algorithm

---

1 **Input**: Finite-armed MAB algorithm Base-Alg satisfying Assumption 1. Subsampling rate $S_m$.
2 **Initialize**: Episode count $\ell \leftarrow 1$, Starting time $t_1^1 \leftarrow 1$.
3 **for** $m = 1, 2, \ldots, \lceil \log(T) \rceil$ **do**
4     Subsample $S_m \wedge 2^m$ arms $\mathcal{A}_m \subset \mathcal{A}$.
5     Initiate a new instance of Base-Alg$(2^m, \mathcal{A}_m)$.
6     **for** $t = t_\ell^m, \ldots, (t_\ell^m + 2^m - 1) \wedge T$ **do**
7         Play arm $a_t$ (receiving reward $Y_t(a_t)$) as chosen by Base-Alg$(2^m, \mathcal{A}_m)$.
8         **Changepoint Test: if**
$$\sum_{s=t_\ell^m}^{t} \hat{\delta}_s(a_s) \geq C_1 \cdot (|\mathcal{A}_m| \vee 2^{m/2}) \cdot \log^3(T)$$
        **then**
9             Restart: $t_{\ell+1}^1 \leftarrow t + 1$, $\ell \leftarrow \ell + 1$.
10             Return to Algorithm 1 (Restart from $m = 1$).
11         **else if** $t = t_\ell^m + 2^m - 1$ **then**
12             $t_\ell^{m+1} \leftarrow t + 1$ (Start of the $(m + 1)$-th block in the $\ell$-th episode).

---

### 4.3. Blackbox Regret Upper Bound

The main result of this section is that Algorithm 1 attains the optimal regret in terms of number of changes $L$ and total-variation $V$ when $\beta \geq 1$ and matches the state-of-art regret bounds with known $L, V$ for $\beta < 1$ (Kim et al., 2024).

**Theorem 2.** *Under Assumption 1 with $\beta \geq 1$, Algorithm 1 with $S_m := \left\lceil 2^{m \cdot \frac{\beta}{\beta+1}} \right\rceil$ satisfies, w.p. $1 - O(1/T)$:*

$$\mathbf{R}_T \leq \tilde{O}\left( (L+1)^{\frac{1}{\beta+1}} T^{\frac{\beta}{\beta+1}} \wedge (V^{\frac{1}{\beta+2}} T^{\frac{\beta+1}{\beta+2}} + T^{\frac{\beta}{\beta+1}}) \right).$$

*If $\beta < 1$, Algorithm 1 with $S_m := \left\lceil 2^{m \cdot \beta/2} \right\rceil$ satisfies w.p. $1 - O(1/T)$:*

$$\mathbf{R}_T \leq \tilde{O}\left( \sqrt{(L+1) \cdot T} \wedge (V^{1/3} T^{2/3} + \sqrt{T}) \right).$$

*Proof.* (Outline) We given an outline of the proof with full details deferred to Section A. We also focus on the setting of $\beta \geq 1$ with (minor) modifications of the argument for $\beta < 1$ discussed in Subsection A.6. Let $t_\ell := t_\ell^1$ be the start of the $\ell$-th episode $[t_\ell, t_{\ell+1})$. Let $\hat{L}$ be the (random) number of restarts triggered over $T$ rounds. Let $m_\ell$ be the index of the last block in $\ell$-th episode.

• **Converting Empirical Regret Bound to Per-Episode Regret Bound.** Following the discussion of Subsection 4.2, we first use concentration to upper bound the per-block regret $\sum_{s=t_\ell^m}^{t_\ell^{m+1}-1} \delta_s(a_s)$ on each block and to also lower bound it on blocks concluding with a restart. We first have by Freedman's inequality (Theorem 7) and AM-GM, with high probability, for all subintervals $[s_1, s_2]$ of rounds:

$$\left| \sum_{s=s_1}^{s_2} \delta_s(a_s) - \hat{\delta}_s(a_s) \right| \lesssim \sqrt{\log(T) \sum_{s=s_1}^{s_2} \delta_s(a_s) + \log(T)}$$

$$\lesssim \frac{1}{2} \sum_{s=s_1}^{s_2} \delta_s(a_s) + \log(T) \qquad (2)$$

This allows us to upper bound the regret on each block $[t_\ell^m, t_\ell^{m+1})$ by $\tilde{O}(S_m)$ using the bound on empirical regret $\sum_{s=t_\ell^m}^{t_\ell^{m+1}-1} \hat{\delta}_s(a_s)$ from Algorithm 1 of Algorithm 1.

Then, summing $S_m \propto 2^{m \cdot \frac{\beta}{\beta+1}}$ over blocks and episodes yields a total regret bound of $\tilde{O}\left( \sum_{\ell=1}^{\hat{L}} (t_{\ell+1} - t_\ell)^{\frac{\beta}{\beta+1}} \right)$.

• **Bounding the Variation Over Each Episode.** It now remains to relate $\sum_{\ell=1}^{\hat{L}} (t_{\ell+1} - t_\ell)^{\frac{\beta}{\beta+1}}$ to the total count $L$ of changes and total-variation $V$. To this end, we show there is a minimal amount of variation in each episode $[t_\ell, t_{\ell+1})$ which will allow us to conclude the total regret bound using arguments similar to prior works (Suk and Kpotufe, 2022, Corollary 2) (Chen et al., 2019, Lemma 5).

We first introduce a regret decomposition, alluded to earlier in (1), based on the "best initially subsampled arm", $\hat{a}_{\ell,m_\ell} := \arg\max_{a \in \mathcal{A}_{m_\ell}} \mu_0(a)$, where we use $\mu_0(a)$ to denote the initial reward of arm $a$ when it's first sampled. The regret in the last block $[t_\ell^{m_\ell}, t_{\ell+1},)$ of the $\ell$-th episode is:

$$\sum_{t=t_\ell^{m_\ell}}^{t_{\ell+1}-1} \delta_t(a_t) = \underbrace{\sum_{t=t_\ell^{m_\ell}}^{t_{\ell+1}-1} \delta_t(\hat{a}_{\ell,m_\ell})}_{(A)} + \underbrace{\sum_{t=t_\ell^{m_\ell}}^{t_{\ell+1}-1} \delta_t(\hat{a}_{\ell,m_\ell}, a_t)}_{(B)}$$

Then from the above and concentration, one of two possible cases must hold if a restart occurs on round $t_{\ell+1}$: either (A) or (B) is $\Omega(S_{m_\ell})$. In either case, we claim large variation occurs over the episode:

$$\sum_{t=t_\ell}^{t_{\ell+1}-1} |\mu_t(a_{t-1}) - \mu_{t-1}(a_{t-1})| \geq (t_{\ell+1} - t_\ell)^{-\frac{1}{\beta+1}}. \quad (3)$$

• **Regret of Best Subsampled Arm is Large.** In the case where (A) is $\Omega(S_{m_\ell})$, we know due to our subsampling rate $S_{m_\ell}$ that $\hat{a}_{\ell,m_\ell}$ will w.h.p. have an initial gap of $\tilde{O}(2^{-m_\ell \cdot \frac{1}{\beta+1}})$ (Theorem 11). On the other hand, (A) being

large means there's a round $t' \in [t_\ell^m, t_{\ell+1})$ such that

$$\delta_{t'}(\hat{a}_{\ell,m_\ell}) \gtrsim S_{m_\ell}/(t_{\ell+1} - t_\ell^{m_\ell}) \gtrsim 2^{-m_\ell \frac{1}{\beta+1}}.$$

Thus, from $2^{-m_\ell \frac{1}{\beta+1}} \geq (t_{\ell+1} - t_\ell)^{-\frac{1}{\beta+1}}$, (3) holds.

• **Regret of Base is Large.** Now, if (B) is $\Omega(S_{m_\ell})$ but (A) is $o(S_{m_\ell})$, suppose for contradiction that (3) is reversed. Then, this means the finite MAB environment experienced by the base algorithm is $(t_{\ell+1} - t_\ell)^{-\frac{1}{\beta+1}}$-mildly corrupt (Theorem 1). Thus, Assumption 2 bounds the regret of the base, which in turn bounds the per-block regret above:

$$\sum_{s=t_\ell^{m_\ell}}^{t_{\ell+1}-1} \delta_t(\hat{a}_{\ell,m_\ell}, a_t) \lesssim t^{\frac{1}{\beta+1}}$$

$$+ \sum_{i=2}^{S_{m_\ell}} \frac{\log(T)}{\Delta_{(i)}} \mathbf{1}\left\{ \frac{\Delta_{(i)}}{4} \geq t^{-\frac{\beta}{\beta+1}} \right\}. \quad (4)$$

where $\{\Delta_{(i)}\}_{i=1}^{S_{m_\ell}}$ are the ordered initial gaps to $\hat{a}_{\ell,m_\ell}$ of the arms in $\mathcal{A}_{m_\ell}$.

We then bound the RHS of (4) by $O(S_{m_\ell})$ to contradict our premise that (B) is $\Omega(S_{m_\ell})$. In Bayati et al. (2020, Lemma D.4), bounding (4) by $O(S_{m_\ell})$ was done in expectation by carefully integrating the densities of each random variable $\Delta_{(i)}^{-1}$. But, this requires additional regularity conditions on an assumed density for the reservoir distribution when $\beta \neq 1$ (op. cit. Section 6.1). To contrast, we bound (4) in high probability using a novel binning argument on the values of the $\Delta_{(i)}$'s and then using Freedman-type concentration on the number of subsampled arms within each bin. Details of this can be found in Subsection A.4. $\qquad \square$

### 4.4. Comparison with MASTER of Wei and Luo (2021)

While our blackbox (Algorithm 1) bears similarities to the MASTER algorithm of Wei and Luo (2021), we highlight that Algorithm 1's design is much simpler in not requiring randomized multi-scale scheduling of re-exploration for the sake of changepoint detection.

The method of detecting changes is also different. MASTER detects changes by comparing the UCB indices of base algorithms scheduled at different times. To contrast, we only track the estimated cumulative regret in each block to track changes. This results in a simpler regret analysis where we don't need to bound the regret incurred while waiting for a re-exploration phase to be triggered.

## 5. Tracking Significant Shifts

While Algorithm 1 is a flexible blackbox which can make use of any reasonable finite-armed MAB algorithm, the

regret bound of Theorem 2 is suboptimal for $\beta < 1$. Futhermore, we aim to show regret upper bounds in terms of the tighter rotting measures of non-stationarity $L_R$ and $V_R$ (cf. Subsection 2.2).

In fact, we go beyond this aim and define a new measure of non-stationarity which precisely tracks the rotting changes which are most severe to performance.

### 5.1. Defining a Significant Shift

Recent works on non-stationary finite-armed MAB achieve regret bounds in terms of more nuanced non-stationarity measures, which only track the most *significant* switches in best arm, or those truly necessitating re-exploration (Suk and Kpotufe, 2022; Buening and Saha, 2023; Suk and Agarwal, 2023; Suk and Kpotufe, 2023; Suk, 2024).

Here, we develop and study an analogous notion for the infinite-armed bandit. First, we note in the infinite-armed bandit problem, there's no single "best arm", as the arm-space is infinite and, almost surely, no sampled arm will have the optimal reward value of 1. Yet, the core inution behind the notion of significant shift for finite MAB (Suk and Kpotufe, 2022, Definition 1) remains by definining a significant shift in terms of the regret experienced by each arm. First, we say an arm $a$ is *safe* on an interval $[s_1, s_2]$ of rounds if:

$$\sum_{s=s_1}^{s_2} 1 - \mu_s(a) \le \kappa_1^{-1} \cdot (s_2 - s_1 + 1)^{\frac{\beta}{\beta+1}}. \qquad (5)$$

Then, a significant shift is roughly defined as occurring when every arm in some set of arms $\hat{\mathcal{A}}$ is unsafe and violates (5). If we let $\hat{\mathcal{A}} = \mathcal{A}$ be the full reservoir set of arms, then this proposed definition may not by meaningful as there could always exist an unsampled arm which is safe in terms of regret, but unknown to the agent.

For the subsampling strategy discussed in Subsection 4.1, we argue it is sensible to let $\hat{\mathcal{A}} = \mathcal{A}_0$, the set of sub-sampled arms. Based on the discussion of Section 4, we see that a subsample of size $\Omega(t^{\frac{\beta}{\beta+1}})$ contains w.h.p. a safe arm with gap $O(t^{-\frac{1}{\beta+1}})$. This motivates the following definition of significant shift, which is defined for any agent (regardless of whether a subsampling strategy is used).

**Definition 3** (Significant Shift)**.** *We say a bandit environment over $t$ rounds is* **safe** *if there exists an arm $a$ among the first $t^{\frac{\beta}{\beta+1}}$ arms sampled by the agent such that (5) holds for all intervals of rounds $[s_1, s_2] \subseteq [t]$. Let $\tau_0 := 1$. Define the $(i+1)$-th* **significant shift** *$\tau_{i+1}$ given $\tau_i$ as the first round $t > \tau_i$ such that the bandit environment over rounds $[\tau_i, t]$ is not safe. Let $\tilde{L}$ be the largest significant shift over $T$ rounds and by convention let $\tau_{\tilde{L}+1} := T + 1$.*

### 5.2. Comparing Significant Shifts with $L_R, V_R$

We first note that the significant shift, like $L_R$ and $V_R$, is an agent-based measure of non-stationarity and so depends on the agent's past decisions.

Next, we caution that a significant shift does not always measure non-stationarity and can in fact be triggered in stationary environments for a "bad" algorithm. For example, consider an agent sampling exactly one arm from the reservoir and committing to it. Then, due to the large variance of a single sample, such an algorithm incurs constant regret with constant probability. In this case, a significant shift is triggered even in a stationary environment as none of the subsample is safe. Thus, in this example, the significant shift tracks the agent's suboptimality.

In spite of this example, we argue that Theorem 3 is still a meaningful measure of non-stationarity for all algorithms of theoretical interest, and indeed learning significant shifts via Algorithm 2 will allow us to attain optimal rates w.r.t. $L_R, V_R$ (Theorem 5). Note that any algorithm attaining the optimal high-probability regret of $O(T^{\frac{\beta}{\beta+1}})$ in stationary environments must sample at least $T^{\frac{\beta}{\beta+1}}$ arms. Intuitively, this is because securing an arm with gap at most $\delta := T^{-\frac{1}{\beta+1}}$ (which occurs for a given arm with probability $\mathbb{P}(\mu_0(a) > 1-\delta) = \Theta(\delta^{\beta})$) requires $\Omega(\delta^{-\beta})$ trials. Indeed, this claim is also seen in the proofs of the stationary regret lower bounds (e.g., Wang et al., 2008, Theorem 3).

At the same time, any sample of $\Omega(T^{\frac{\beta}{\beta+1}})$ arms from the reservoir will contain an arm with gap $O(\delta)$ w.h.p.. (Theorem 11), which will maintain (5) over all subintervals. Then, a significant shift will not be triggered unless this best sub-sampled arm' gap increases. Thus, any algorithm attaining optimal regret in stationary environments will satisfy w.h.p. $\tilde{L} \le L$.

Additionally, the only way an arm with initial gap $O(\delta)$ becomes unsafe and violates (5) is if there's large rotting total variation (i.e., $V_R$ is bounded below). This means $\tilde{L} \le L_R$ and we'll see in the next subsection that learning significant shifts in fact allows us to recover the optimal rate $V_R^{\frac{1}{\beta+2}} \cdot T^{\frac{\beta+1}{\beta+2}}$ in terms of $V_R$ (Theorem 5).

### 5.3. Regret Upper Bounds

Here, we derive a tight and adaptive regret bound in terms of the significant shifts. In particular, we show a regret bound of $(\tilde{L}+1)^{\frac{1}{\beta+1}} T^{\frac{\beta}{\beta+1}} \wedge (V_R^{\frac{1}{\beta+2}} T^{\frac{\beta+1}{\beta+2}} + T^{\frac{\beta}{\beta+1}})$ in terms of $\tilde{L}$ significant shifts and $V_R$ total rotting variation.

A preliminary task here is as follows:

**Goal 1.** *Show a regret bound of $\tilde{O}(t^{\frac{\beta}{\beta+1}})$ in $t$-round safe environments.*
Given a base procedure achieving said goal, we can then use

a restart strategy similar to Algorithm 1 where we restart the base upon detecting a significant shift.

For the finite $K$-armed setting, the analogous preliminary claim (Suk, 2024, Theorem 11) is to show a regret bound of $O(\sqrt{KT})$ in safe environments. This is achieved using a randomized variant of successive elimination (Even-Dar et al., 2006).

However, once again the infinite-armed setting is more nuanced and so this claim cannot directly be applied. Indeed, setting $K := T^{\frac{\beta}{\beta+1}}$ results in a suboptimal rate of $\sqrt{KT} = T^{\frac{\beta+1/2}{\beta+1}}$. The fundamental issue here is that the $\sqrt{KT}$ rate captures a worst-case variance of estimating bounded rewards. To get around this, we do a more refined variance-aware regret analysis relying on self-bounding techniques similar to those used to show Theorem 2 in Section 4.

To further emphasize the more challenging nature of showing Goal 1, we notice that the regret analysis of our blackbox procedure Algorithm 1 in Subsection 4.3 crucially relies on bounding the logarithmic gap-dependent regret rate of Assumption 2 over periods of small total-variation (used to show (3)). However, such a gap-dependent regret rate is ill-defined when the gaps $\Delta_i$ can be changing substantially over time (as to violate Assumption 2) which can happen in safe environments while we expect stationary regret rates.

Nevertheless, we show that a different per-arm regret analysis gets around this difficulty. Our procedure (Algorithm 2) is a restarting randomized elimination using the same doubling block scheme of Algorithm 1.

Going into more detail, we note, by uniformly exploring a *candidate armset* $\mathcal{G}_t$ at round $t$, we can maintain importance-weighted estimates of the gaps of each arm $a \in \mathcal{G}_t$:

$$\hat{\delta}_t^{\mathrm{IW}}(a) := \frac{(1 - Y_t(a_t)) \cdot \mathbf{1}\{a_t = a\}}{\mathbb{P}(a_t = a \mid \mathcal{H}_{t-1})},$$

where $\mathcal{H}_{t-1}$ is the $\sigma$-algebra generated by decisions and observations up to round $t - 1$.

Next, we note, by Freedman's inequality that the estimation error of the cumulative estimate $\sum_{t \in I} \hat{\delta}_t^{\mathrm{IW}}(a)$ over an interval $I$ of rounds scales like $\sqrt{\sum_{t \in I} \delta_t(a) \cdot |\mathcal{G}_t|}$. The inclusion of the $\delta_t(a)$ term inside of the square root is crucial here and, using self-bounding arguments, yield tighter concentration bounds of order $\tilde{O}(\max_{t \in I} |\mathcal{G}_t|)$, which we can use as a threshold for fast variance-based elimination.

**Theorem 4.** *Let $\hat{L}$ be the number of episodes $[t_\ell, t_{\ell+1})$ elapsed in Algorithm 2 over $T$ rounds. Algorithm 2 satisfies, w. p. at least $1 - 1/T$:*

$$\mathbf{R}_T = \tilde{O}\left(\sum_{\ell=1}^{\hat{L}} (t_{\ell+1} - t_\ell)^{\frac{\beta}{\beta+1}}\right).$$

---

**Algorithm 2:** Restarting Subsampling Elimination

1 **Initialize**: Episode count $\ell \leftarrow 1$, start $t_1^1 \leftarrow 1$.
2 **for** $m = 1, 2, \ldots, \lceil \log(T) \rceil$ **do**
3    Subsample $\left\lceil 2^{(m+1) \cdot \frac{\beta}{\beta+1}} \log(T) \right\rceil \wedge 2^m$ arms $\mathcal{A}_m$ and let $\mathcal{G}_{t_\ell^m} \leftarrow \mathcal{A}_m$.
4    **for** $t = t_\ell^m, \ldots, (t_\ell^m + 2^m - 1) \wedge T$ **do**
5      Play arm $a_t$ as $\mathrm{Unif}\{\mathcal{G}_t\}$ and observe reward $Y_t(a_t)$.
6      Eliminate arms: $\mathcal{G}_{t+1} \leftarrow$
$$\mathcal{G}_t \setminus \left\{ a : \sum_{s=t_\ell^m}^{t} \hat{\delta}_s^{\mathrm{IW}}(a) \geq C_2 \cdot |\mathcal{A}_m| \log(T) \right\}.$$
7      **Restart Test: if** $\mathcal{G}_{t+1} = \emptyset$ **then**
8        Restart: $t_{\ell+1}^1 \leftarrow t + 1, \ell \leftarrow \ell + 1$.
9        Return to Algorithm 2 (Restart from $m = 1$).
10      **else if** $t = t_\ell^m + 2^m - 1$ **then**
11        $t_\ell^{m+1} \leftarrow t + 1$ (Start of the $m + 1$-th *block* in the $\ell$-th episode).

---

*Proof.* (Outline) We give a proof outline here and full details are found in Section B. It suffices to bound the regret on block $[t_\ell^m, t_\ell^{m+1})$ by $\tilde{O}(S_m)$. To show this, we do a variance-aware version of the per-arm regret analysis of Section B.1 in Suk and Kpotufe (2022).

We first transform the regret according to its conditional expectation. Note that, from the uniform sampling strategy,

$$\mathbb{E}[\delta_t(a_t) \mid \mathcal{H}_{t-1}] = \sum_{a \in \mathcal{G}_t} \frac{\delta_t(a)}{|\mathcal{G}_t|}.$$

Next, note that

$$\mathrm{Var}[\delta_t(a_t) \mid \mathcal{H}_{t-1}] \leq \mathbb{E}[\delta_t^2(a_t) \mid \mathcal{H}_{t-1}] \leq \mathbb{E}[\delta_t(a_t) \mid \mathcal{H}_{t-1}].$$

Then, using Freedman's inequality (Theorem 7) with the above, we have for all subintervals $I \subseteq [T]$, with probability at least $1 - 1/T$:

$$\sum_{t \in I} \delta_t(a_t) - \sum_{a \in \mathcal{G}_t} \frac{\delta_t(a)}{|\mathcal{G}_t|} \lesssim \sqrt{\log(T) \sum_{t \in I} \mathbb{E}[\delta_t(a_t) \mid \mathcal{H}_{t-1}]}$$
$$+ \log(T)$$
$$\lesssim \sum_{t \in I} \sum_{a \in \mathcal{G}_t} \frac{\delta_t(a)}{|\mathcal{G}_t|} + \log(T),$$

where the second inequality is from AM-GM. In light of the above, it remains to bound $\sum_{a=1}^{K} \sum_{t=t_\ell^m}^{t^a} \frac{\delta_t(a)}{|\mathcal{G}_t|}$ where $t^a$ is the last round in block $[t_\ell^m, t_\ell^{m+1})$ that $a$ is retained.

We next again use Freedman's inequality (Theorem 7) and self-bounding to relate $\sum_{t \in I} \delta_t(a)$ to $\sum_{t \in I} \hat{\delta}_t^{\mathrm{IW}}(a)$. We

have w.p. at least $1 - 1/T$:

$$\left| \sum_{t=t_\ell^m}^{t^a} \delta_t(a) - \hat{\delta}_t^{\text{IW}}(a) \right|$$

$$\lesssim \max_{t \in [t_\ell^m, t_a]} |\mathcal{G}_t| \log(T) + \sqrt{\log(T) \sum_{t=t_\ell^m}^{t^a} \delta_t(a) \cdot |\mathcal{G}_t|}$$

$$\leq \frac{1}{2} \sum_{t=t_\ell^m}^{t^a} \delta_t(a) + C' \max_{t \in [t_\ell^m, t^a]} |\mathcal{G}_t| \log(T), \qquad (6)$$

where again we use AM-GM in the second inequality. Next,

$$\sum_{t=t_\ell^m}^{t^a} \frac{\delta_t(a)}{|\mathcal{G}_t|} \leq \left( \max_{s \in [t_\ell^m, t^a]} \frac{1}{|\mathcal{G}_s|} \right) \sum_{t=t_\ell^m}^{t^a} \delta_t(a).$$

Moving the $\frac{1}{2} \sum_{t \in I} \delta_t(a)$ to the other side in (6), we get

$$\sum_{t=t_\ell^m}^{t^a} \delta_t(a) \lesssim \sum_{t=t_\ell^m}^{t^a} \hat{\delta}_t^{\text{IW}}(a) + \log(T) \max_{t \in [t_\ell^m, t^a]} |\mathcal{G}_t|.$$

Combining the above two displays with Algorithm 2 of Algorithm 2, we have:

$$\sum_{t=t_\ell^m}^{t^a - 1} \frac{\delta_t(a)}{|\mathcal{G}_t|} \lesssim \max_{s \in [t_\ell^m, t^a - 1]} \frac{S_m}{|\mathcal{G}_s|} \log(T),$$

Now, summing $\max_{s \in [t_\ell^m, t^a-1]} |\mathcal{G}_s|^{-1}$ over arms $a \in \mathcal{A}_m$ yields another $\log(S_m)$ factor, while summing over blocks $m \in [m_\ell]$ and episodes $\ell \in [\hat{L}]$ finishes the proof. $\qquad \square$

We next show the regret bound of Theorem 4 in fact recovers the minimax regret rates in terms of $\tilde{L}$ and $V_T$.

**Corollary 5.** *Algorithm 2 satisfies, w.p. at least $1 - 1/T$:*

$$\mathbf{R}_T \leq \tilde{O}\left( (\tilde{L} + 1)^{\frac{1}{\beta+1}} T^{\frac{\beta}{\beta+1}} \wedge (V_R^{\frac{1}{\beta+2}} T^{\frac{\beta+1}{\beta+2}} + T^{\frac{\beta}{\beta+1}}) \right).$$

*Proof.* (Sketch) Using similar arguments to the proof of Theorem 2, within each block the best initial arm $\hat{a}_{\ell,m}$ has gap $\tilde{O}(2^{-m \cdot \frac{1}{\beta+1}})$. On the other hand, within a block $[t_\ell^{m_\ell}, t_{\ell+1})$ ending in a restart, this best initial arm is eliminated meaning there exists a round $t \in [t_\ell^{m_\ell}, t_{\ell+1})$ such that $\delta_t(\hat{a}_{\ell,m_\ell}) \gtrsim (t_{\ell+1} - t_\ell^{m_\ell})^{-\frac{1}{\beta+1}}$. Thus, the gap of $\hat{a}_{\ell,m_\ell}$ must have increased by at least this amount which gives a lower bound on the per-episode total variation of $\Omega((t_{\ell+1} - t_\ell)^{-\frac{1}{\beta+1}})$. Then, by similar arguments to the proof of Theorem 2, we deduce the total regret bound. $\qquad \square$

## 6. Comparing Blackbox vs. Elimination

Theorem 5 and the nearly matching lower bounds of Section 3 show that only rotting non-stationarity ($\tilde{L} \leq L_R$ and $V_R$) factor into the difficulty of non-stationarity. In other words, rising non-stationarity is benign for non-stationary infinite-armed bandits. Intuitively, this is because our problem assumes knowledge of both the top reward value and upper bound on rewards, which coincide and are equal to 1. Hence, arms with rising rewards require less exploration.

Interestingly, unlike the case with Algorithm 1, the elimination algorithm's bound in Theorem 5 does *not* require an upper bound on masses of the reservoir (Assumption 1), or there's no dependence on $\kappa_2$ in the regret upper bounds of Theorem 5. This is new to this work and was not known even in the previous stationary regret bounds (Wang et al., 2008; Bayati et al., 2020; Kim et al., 2024). This suggests our regret analysis is simpler and, indeed, we only require that the initial best arm in the subsample has small gap (which only requires lower bounded masses of the reservoir as we see in Theorem 11). To contrast, the regret analysis of Theorem 2 (more similar to those of the aforementioned works) uses the upper bound on tail probabilities scaling with $\kappa_2$ to bound the regret of the finite-armed MAB base algorithm. Such a step is avoided in the regret analysis of Theorem 4 by estimating the regret of each arm separately.

On the other hand, the blackbox can be seen as more extensible as it allows for a wide range of finite-armed MAB algorithms to be used as a base. We finally note Algorithm 2 can essentially be reformulated as an instantiation of the blackbox with a successive elimination base algorithm.

## 7. Experiments

Here, we demonstrate the performance of Algorithms 1 and 2 on synthetic datasets. For the Base-Alg in Algorithm 1 we use UCB (Auer et al., 2002). As benchmarks, we implement SSUCB (Bayati et al., 2020) and a variant of it using a sliding window with size $\sqrt{T}$ (SSUCB-SW), and AUCBT-ASW (Kim et al., 2024), which achieves a suboptimal regret bound of $\tilde{O}(\min\{V^{\frac{1}{\beta+2}} T^{\frac{\beta+1}{\beta+2}}, (L + 1)^{\frac{1}{\beta+1}} T^{\frac{\beta}{\beta+1}}\} + \min\{T^{\frac{2\beta+1}{2\beta+2}}, T^{\frac{3}{4}}\})$ in rested rotting setups.

• **Comparing Total Variation-based Regret Bounds.** To ensure a fair comparison between algorithms, we consider a rotting scenario where the mean reward of each selected arm decreases at a rate of $O(1/t)$ at time $t$. In this environment, for all our algorithms, it can be shown that $L = \Omega(T)$, $V = \tilde{O}(1)$. For the case of $\beta = 1$ such that initial mean rewards follow a uniform distribution on $[0, 1]$ (Figure 1(b)), both our algorithms outperform the benchmarks, with the elimination one achieving the best performance. These results validate the insights of Section 6. Specifically, our

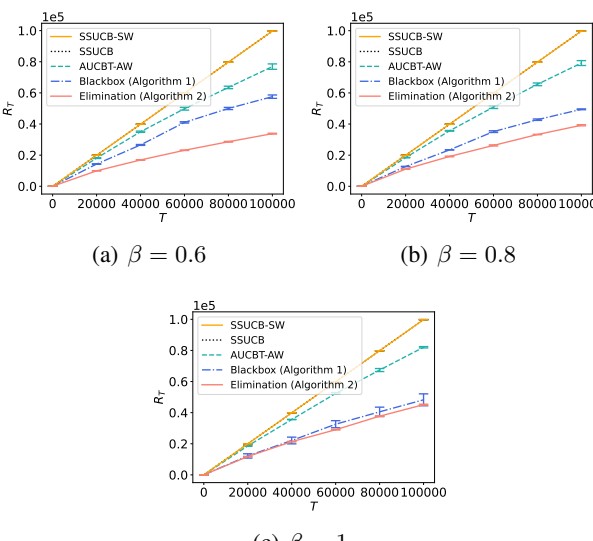

(a) $\beta = 0.6$        (b) $\beta = 0.8$

(c) $\beta = 1$

Figure 1: Experimental results showing regret of algorithms

algorithms have a regret bound of $\tilde{O}(T^{2/3})$ (Theorem 2 and Theorem 5) vs. `AUCBT-ASW`'s bound of $\tilde{O}(T^{3/4})$ (Kim et al., 2024).

In Figure 1, we observe that our algorithms outperform the benchmarks across various $\beta$ values, aligning with theoretical results. Furthermore, the performance gap between the elimination and blackbox algorithms increases as $\beta$ decreases, which is also consistent with our theoretical results.

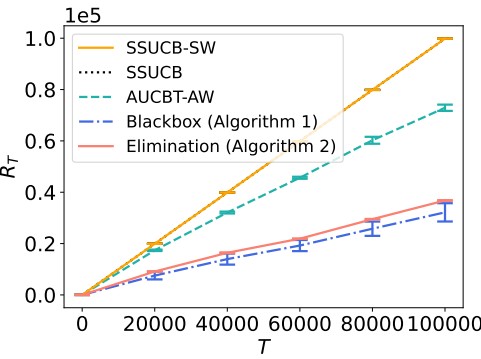

Figure 2: Experimental results showing regret of algorithms

● **Comparing Regret Bounds based on Number of Changes.** Figure 2 covers the piecewise stationary setting where we use a rotting rate of $\rho_t = 1$ at $L = \sqrt{T}$ different rounds with $\beta = 1$. The prior art `AUCBT-ASW` (Kim et al., 2024) has a regret bound of $\sqrt{LT} = T^{3/4}$, yet we see from the plot that our procedures have empirically better regret owing to the small number of significant shifts. Surprisingly, we see the blackbox algorithm has comparable performance to our elimination procedure, suggesting that

the blackbox algorithm may also be capable of adapting to significant shifts.

**Remark 5.** *Our implementation of Algorithm 2 does not include the* $\log(T)$ *factor in the subsampling rate of Algorithm 2, as this lead to more stable experimental results. One can show this only changes the bound of Theorem 5 up to a* $\log^{2/\beta}(T)$ *factor, which is not large for* $\beta \in \{0.6, 0.8, 1\}$.

## 8. Conclusion and Future Directions

We've shown the first optimal and adaptive dynamic regret bounds for infinite-armed non-stationary bandit with arms drawn from a reservoir distribution. For future work, it would be interesting to see if our techniques can be extended to non-stationary linear bandits, and to design more practical/computationally efficient procedures for detecting changes.

## Impact Statement

As our contribution here is primarily theoretical, advancing the state-of-art in infinite-armed non-stationary bandits, we do not foresee any major societal impact concerns.

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

# A. Blackbox Algorithm Regret Analysis (Details for the Proof of Theorem 2)

Here, we present the details of the proof of Theorem 2 following the outline of Section 4. Again, we focus on the case of $\beta \geq 1$ and discuss modifications of the argument required for $\beta < 1$ in Subsection A.6.

## A.1. Preliminaries

Let $c_0, c_1, c_2, \ldots$ denote constants not depending on $T$, $\kappa_1$, or $\kappa_2$ (Assumption 1). In what follows, all logarithms will be base 2. We'll also assume WLOG that $\log(T) \geq \kappa_1^{-\frac{1}{\beta}} \vee \kappa_2$, as otherwise we can bound the regret by a constant only depending on $\kappa_1$ and $\kappa_2$.

Next, we establish a basic fact about the block structure of Algorithm 1.

**Fact 6.** *Recall from Section 4 that $m_\ell$ is the index of the last block in the $\ell$-th episode. Then, for any episode $[t_\ell, t_{\ell+1})$ terminating in a restart (via Algorithm 1 of Algorithm 1), we have*

$$S_{m_\ell} = \left\lceil 2^{m_\ell \cdot \frac{\beta}{\beta+1}} \right\rceil \leq 2^{m_\ell}, \tag{7}$$

*and*

$$(t_{\ell+1} - t_\ell^{m_\ell})^{\frac{1}{\beta+1}} \geq \frac{C_1}{2^{\frac{1}{\beta+1}}} \log^3(T) \tag{8}$$

*Proof.* By Algorithm 1 of Algorithm 1, we must have

$$2^{m_\ell} \geq t_{\ell+1} - t_\ell^{m_\ell} + 1 \geq \sum_{s=t_\ell^{m_\ell}}^{t_{\ell+1}-1} 1 - Y_s(a_s) \geq C_1 \cdot |\mathcal{A}_{m_\ell}| \cdot \log^3(T).$$

Now, recall the definition of $S_m$ so that $|\mathcal{A}_{m_\ell}| := \left\lceil 2^{m \cdot \frac{\beta}{\beta+1}} \right\rceil \wedge 2^m$. We first note that we cannot have $|\mathcal{A}_{m_\ell}| = 2^{m_\ell}$ since the above display would then become $1 \geq C_1 \log^3(T)$. Thus

$$t_{\ell+1} - t_\ell^{m_\ell} + 1 \geq C_1 \cdot 2^{m_\ell \cdot \frac{\beta}{\beta+1}} \cdot \log^3(T) \geq C_1 \cdot (t_{\ell+1} - t_\ell^{m_\ell} + 1)^{\frac{\beta}{\beta+1}} \cdot \log^3(T).$$

Rearranging this becomes $(t_{\ell+1} - t_\ell^{m_\ell} + 1)^{\frac{1}{\beta+1}} \geq C_1 \log^3(T)$. Further, bounding $t_{\ell+1} - t_\ell^{m_\ell} + 1 \leq 2 \cdot (t_{\ell+1} - t_\ell^{m_\ell})$ finishes the proof. $\square$

## A.2. Using Concentration to Bound Per-Block Regret (Proof of Theorem 8)

We first present Freedman's inequality which is used in Section 4 to bound the regret using the empirical regret bound of Algorithm 1 of Algorithm 1.

**Lemma 7** (Strengthened Freedman's Inequality, Theorem 9 (Zimmert and Lattimore, 2022)). *Let $X_1, X_2, \ldots, X_T$ be a martingale difference sequence with respect to a filtration $\mathcal{F}_1 \subseteq \mathcal{F}_2 \subseteq \cdots \subseteq \mathcal{F}_T$ such that $\mathbb{E}[X_t \mid \mathcal{F}_t] = 0$ and assume $\mathbb{E}[|X_t| \mid \mathcal{F}_t] < \infty$ a.s.. Then, with probability at least $1 - \delta$,*

$$\sum_{t=1}^{T} X_t \leq 3\sqrt{V_t \log\left(\frac{2 \max\{U_t, \sqrt{V_T}\}}{\delta}\right)} + 2U_T \log\left(\frac{2\max\{U_T, \sqrt{V_T}\}}{\delta}\right),$$

*where $V_T = \sum_{t=1}^{T} \mathbb{E}[X_t^2 \mid \mathcal{F}_t]$, and $U_T = \max\{1, \max_{s \in [T]} X_s\}$.*

We next use this to relate the per-episode regret to the empirical bounds of Algorithm 1.

**Lemma 8.** *Let $\mathcal{E}_1$ be the event that (a) for all blocks $[t_\ell^m, t_\ell^{m+1})$,*

$$\sum_{s=t_\ell^m}^{t_\ell^{m+1}-1} \delta_s(a_s) < 3C_1 \cdot |\mathcal{A}_m| \cdot \log^3(T). \tag{9}$$

and (b) for the last block $[t_\ell^{m_\ell}, t_{\ell+1})$ of episodes $[t_\ell, t_{\ell+1})$ concluding in a restart, we have:

$$\sum_{s=t_\ell^{m_\ell}}^{t_{\ell+1}-1} \delta_s(a_s) \geq \frac{C_1}{2} \cdot |\mathcal{A}_{m_\ell}| \cdot \log^3(T). \tag{10}$$

Then, $\mathcal{E}_1$ occurs with probability at least $1 - 1/T$.

*Proof.* First, fix an interval of rounds $[s_1, s_2]$. Then, we note that

$$\sum_{s=s_1}^{s_2} 1 - Y_s(a_s) - \mathbb{E}[1 - Y_s(a_s) \mid \mathcal{H}_{s-1}],$$

is a martingale difference sequence with respect to the natural filtration $\{\mathcal{H}_t\}_t$ of $\sigma$-algebras $\mathcal{H}_t$, generated by the observations and decisions (of both decision-maker and adversary) up to round $t - 1$. Now, since the choice of arm $a_t$ at round $t$ is fixed conditional on $\mathcal{H}_{t-1}$, we have $\mathbb{E}[Y_t(a_t) \mid \mathcal{H}_{t-1}] = \mu_t(a_t)$. We also have since $Y_t(a_t) \in [0, 1]$:

$$\text{Var}(1 - Y_t(a_t) \mid \mathcal{H}_{t-1}) \leq \mathbb{E}[(1 - Y_t(a_t))^2 \mid \mathcal{H}_{t-1}] \leq \mathbb{E}[1 - Y_t(a_t) \mid \mathcal{H}_{t-1}] = \delta_t(a_t).$$

Then, by Freedman's inequality (Theorem 7) we have with probability at least $1 - 1/T^3$: for all choices of intervals $[s_1, s_2] \subseteq [T]$:

$$\left| \sum_{s=s_1}^{s_2} (1 - Y_s(a_s)) - (1 - \mu_s(a_s)) \right| \leq 3\sqrt{\log(2T^3) \sum_{s=s_1}^{s_2} \delta_s(a_s) + 2\log(2T^3)}$$

$$\leq \frac{1}{2} \sum_{s=s_1}^{s_2} \delta_s(a_s) + \frac{13}{2} \log(2T^3), \tag{11}$$

where the second inequality is by AM-GM. Going forward, suppose the above concentration holds for all subintervals of rounds $[s_1, s_2] \subseteq [T]$.

At the same time, on each $m$-th block, Algorithm 1 of Algorithm 1 gives us an empirical regret upper bound since the changepoint test is not triggered until possibly the last round of the block. Specifically, the inequality in Algorithm 1 of Algorithm 1 must be reversed for the second-to-last round of the block or

$$\sum_{s=t_\ell^m}^{t_\ell^{m+1}-2} 1 - Y_s(a_s) < C_1 \cdot (|\mathcal{A}_m| \vee 2^{m/2}) \cdot \log^3(T),$$

where by convention we let $t_\ell^{m_\ell+1} := t_{\ell+1}$. Thus,

$$\sum_{s=t_\ell^m}^{t_\ell^{m+1}-1} 1 - Y_s(a_s) < C_1 \cdot (|\mathcal{A}_m| \vee 2^{m/2}) \cdot \log^3(T) + 1. \tag{12}$$

Combining the above with (11), we conclude

$$\sum_{s=t_\ell^m}^{t_\ell^{m+1}-1} \delta_s(a_s) < 2C_1 \cdot (|\mathcal{A}_m| \vee 2^{m/2}) \cdot \log^3(T) + 2 + 13\log(2T^3) \leq 3C_1 \cdot (|\mathcal{A}_m| \vee 2^{m/2}) \cdot \log^3(T).$$

where the last inequality holds for $C_1$ large enough.

At the same time, for constant $C_1$ in Algorithm 1 of Algorithm 1 chosen large enough, we have that, for the last block $[t_\ell^{m_\ell}, t_{\ell+1})$ of an episode concluding in a restart, we must have

$$\sum_{s=t_\ell^{m_\ell}}^{t_{\ell+1}-1} 1 - Y_s(a_s) \geq C_1 \cdot (|\mathcal{A}_m| \vee 2^{m/2}) \cdot \log^3(T) \implies$$

$$\sum_{s=t_\ell^{m_\ell}}^{t_{\ell+1}-1} \delta_s(a_s) \geq \frac{2C_1}{3} \cdot (|\mathcal{A}_{m_\ell}| \vee 2^{m/2}) \cdot \log^3(T) - \frac{13}{3}\log(2T^3) \geq \frac{C_1}{2} \cdot (|\mathcal{A}_{m_\ell}| \vee 2^{m/2}) \cdot \log^3(T),$$

where the first inequality in the second line above comes from combining the first line with (11) and the last inequality holds for $C_1$ large enough.

Finally, we note that for $\beta \geq 1$, we have $|\mathcal{A}_m| = S_m \wedge 2^m = \left\lceil 2^{m \cdot \frac{\beta}{\beta+1}} \right\rceil \wedge 2^m \geq 2^{m/2}$. Thus, $|\mathcal{A}_m| \vee 2^{m/2} = |\mathcal{A}_m|$ in all our above inequalities. $\qquad\square$

### A.3. Summing Regret Over Blocks

Next, we sum the per-block regret bound of Theorem 8 over blocks $m$ and episodes $\ell$ to obtain a total regret bound.

**Lemma 9.** *Under event $\mathcal{E}_1$, we have*

$$\sum_{t=1}^{T} \delta_t(a_t) \leq c_0 \log^3(T) \sum_{\ell \in [\hat{L}]} (t_{\ell+1} - t_\ell)^{\frac{\beta}{\beta+1}}.$$

We first decompose the regret along episodes and blocks contained therein:

$$\sum_{t=1}^{T} 1 - \mu_t(a_t) = \sum_{\ell=1}^{\hat{L}} \sum_{m \in [m_\ell]} \sum_{t=t_\ell^m}^{t_\ell^{m+1}-1} 1 - \mu_t(a_t).$$

Then, on event $\mathcal{E}_1$, we have summing (9):

$$\sum_{t=1}^{T} 1 - \mu_t(a_t) \leq \sum_{\ell \in [\hat{L}]} \sum_{m \in [m_\ell]} 3C_1 \cdot |\mathcal{A}_m| \cdot \log^3(T)$$

$$\leq c_1 \sum_{\ell \in [\hat{L}]} \sum_{m \in [m_\ell]} 2^{m \cdot \frac{\beta}{\beta+1}} \log^3(T)$$

$$\leq c_2 \sum_{\ell \in [\hat{L}]} (t_{\ell+1} - t_\ell)^{\frac{\beta}{\beta+1}} \cdot \log^3(T).$$

where in the last line we sum the geometric series over $m$ and the fact that $2^{m_\ell \cdot \frac{\beta}{\beta+1}} \leq (t_{\ell+1} - t_\ell)^{\frac{\beta}{\beta+1}} \wedge 2^{m_\ell}$.

### A.4. Showing there is Large Variation in Each Episode

Next, following the proof outline of Section 4, our goal is to show there is a minimal amount of variation in each episode.

**Lemma 10.** $\forall \ell \in [\hat{L} - 1]$, *w.p.* $\geq 1 - 4/T$:

$$\sum_{t=t_\ell}^{t_{\ell+1}-1} |\mu_t(a_{t-1}) - \mu_{t-1}(a_{t-1})| \geq \frac{\log^3(T)}{(t_{\ell+1} - t_\ell)^{\frac{1}{\beta+1}}}.$$

*Proof.* As outlined in the proof outline of Section 4, in light of the regret lower bound (10) of Theorem 8, we consider two

different cases which we recall below:

$$\sum_{t=t_\ell^{m_\ell}}^{t_{\ell+1}-1} 1 - \mu_t(\hat{a}_{\ell,m_\ell}) \geq \frac{C_1}{4}|\mathcal{A}_{m_\ell}|\log^3(T) \tag{A}$$

$$\sum_{t=t_\ell^{m_\ell}}^{t_{\ell+1}-1} 1 - \mu_t(\hat{a}_{\ell,m_\ell}) \leq \frac{C_1}{4}|\mathcal{A}_{m_\ell}|\log^3(T) \quad \text{and} \quad \sum_{t=t_\ell^{m_\ell}}^{t_{\ell+1}-1} \mu_t(\hat{a}_{\ell,m_\ell}) - \mu_t(a_t) \geq \frac{C_1}{4}|\mathcal{A}_{m_\ell}|\log^3(T) \tag{B}$$

Now, on event $\mathcal{E}_1$, due to (10), one of (A) or (B) must hold. Our goal is to show that in either case, large variation must have elapsed over the episode.

● **Best Initial Arm has Large Dynamic Regret.** We first consider the former case (A). First, we establish a lemma asserting the best initially subsampled arm has small initial gap with high probability.

**Lemma 11.** *(Proof in Subsection A.7) Recall that $\mu_0(a)$ is the initial mean reward of arm $a$. Let $\hat{a}_{\ell,m_\ell}$ denote the best initial arm among the arms sampled in the last block of episode $[t_\ell, t_{\ell+1})$ or $\hat{a}_{\ell,m_\ell} := \arg\max_{a \in \mathcal{A}_{m_\ell}} \mu_0(a)$. Let $\mathcal{E}_3$ be the event that*

$$\forall \ell \in [\hat{L}] : 1 - \mu_0(\hat{a}_{\ell,m_\ell}) \leq \frac{\log^3(T)}{2^{m_\ell \cdot \frac{1}{\beta+1}}}.$$

*Then, $\mathcal{E}_3$ occurs with probability at least $1 - 1/T$.*

Now, from (A) (with large enough $C_1 > 16$), we also know that there exists a round $t' \in [t_\ell^{m_\ell}, t_{\ell+1})$ such that

$$1 - \mu_{t'}(\hat{a}_{\ell,m_\ell}) > \frac{C_1 \cdot |\mathcal{A}_{m_\ell}| \cdot \log^3(T)}{4(t_{\ell+1} - t_\ell^{m_\ell})} \geq \frac{16 \cdot 2^{m_\ell \cdot \frac{\beta}{\beta+1}} \cdot \log^3(T)}{4 \cdot 2 \cdot 2^{m_\ell}} \geq \frac{2\log^3(T)}{2^{m_\ell \frac{1}{\beta+1}}}.$$

This means the mean reward of arm $\hat{a}_{\ell,m_\ell}$ must have moved by amount at least $2^{-m_\ell \cdot \frac{1}{\beta+1}}\log^3(T)$ over the course of the block, implying

$$\mu_0(\hat{a}_{\ell,m_\ell}) - \mu_{t'}(\hat{a}_{\ell,m_\ell}) \geq \frac{\log^3(T)}{2^{m_\ell \cdot \frac{1}{\beta+1}}} \geq \frac{\log^3(T)}{(t_{\ell+1} - t_\ell)^{\frac{1}{\beta+1}}}.$$

Since the adversary can only modify the rewards of an arm $a_t$ after the round $t$ it is played, the movement must have occurred on rounds where $\hat{a}_{\ell,m_\ell}$ was chosen by the agent. Thus,

$$\sum_{t=t_\ell}^{t_{\ell+1}-1} |\mu_t(a_{t-1}) - \mu_{t-1}(a_{t-1})| \geq \sum_{t=t_\ell}^{t_{\ell+1}-1} |\mu_t(\hat{a}_{\ell,m_\ell}) - \mu_{t-1}(\hat{a}_{\ell,m_\ell})| \geq \mu_0(\hat{a}_{\ell,m_\ell}) - \mu_{t'}(\hat{a}_{\ell,m_\ell}) \geq \frac{\log^3(T)}{(t_{\ell+1} - t_\ell)^{\frac{1}{\beta+1}}}.$$

● **Regret of Base Algorithm is Large.** Now, we consider the other case (B), where the regret of the subsampled bandit environment over the rounds $[t_\ell^{m_\ell}, t_{\ell+1})$ is large, while the dynamic regret of the best initial arm is small. In this case, we want to show a contradiction: that if the total variation over the episode is small, then the finite MAB environment experienced by the base algorithm must be mildly corrupt (Theorem 1) which means Assumption 2 can be used to bound the regret of the last block. This will yield a contradiction since by virtue of Algorithm 1 being triggered, we know the regret of the last block be large.

Suppose for contradiction that

$$\sum_{t=t_\ell}^{t_{\ell+1}-1} |\mu_t(a_{t-1}) - \mu_{t-1}(a_{t-1})| < \frac{\log^3(T)}{(t_{\ell+1} - t_\ell)^{\frac{1}{\beta+1}}}. \tag{13}$$

Then, since our adaptive adversary only changes the rewards of arms on the rounds they are played, we have the finite-armed MAB environment experienced by the base is $\alpha$-mildly corrupt (Theorem 1) with respect to reference reward profile $\{\mu(a)\}_{a \in \mathcal{A}_{m_\ell}}$ and $\alpha := \frac{\log^3(T)}{(t_{\ell+1} - t_\ell^{m_\ell})^{\frac{1}{\beta+1}}}$.

This means we can employ Assumption 2 to bound the regret. In particular, our end goal here is to show that

$$\sum_{t=t_\ell^m}^{t_{\ell+1}-1} \mu_t(\hat{a}_{\ell,m_\ell}) - \mu_t(a_t) < \frac{C_1}{4} \cdot |\mathcal{A}_{m_\ell}| \cdot \log^3(T), \tag{14}$$

which will contradict (B) and imply (13) is true. First, by Assumption 2 with $\alpha = \frac{\log^3(T)}{(t_{\ell+1}-t_\ell^{m_\ell})^{\frac{1}{\beta+1}}}$, we have

$$\sum_{t=t_\ell^{m_\ell}}^{t_{\ell+1}-1} \mu_t(\hat{a}_{\ell,m_\ell}) - \mu_t(a_t) \le C_0 \left( \sum_{i=2}^{|\mathcal{A}_{m_\ell}|} \frac{\log(T)}{\Delta_{(i)}} \cdot \mathbf{1}\left\{ \frac{\Delta_{(i)}}{4} \ge \frac{\log^3(T)}{(t_{\ell+1}-t_\ell^{m_\ell})^{\frac{1}{\beta+1}}} \right\} + (t_{\ell+1}-t_\ell^{m_\ell})^{\frac{\beta}{\beta+1}} \cdot \log^3(T) \right).$$

To ease notation, we now parametrize the arms in $\mathcal{A}_{m_\ell} \backslash \{\hat{a}_{\ell,m_\ell}\}$ via $\{2, \ldots, |\mathcal{A}_{m_\ell}|\}$ and let $\Delta_i$ be the initial gap of arm $i$ to $\hat{a}_{\ell,m_\ell}$:

$$\Delta_i := \mu_0(\hat{a}_{\ell,m_\ell}) - \mu_0(i),$$

Now, we will partition the values of $\Delta_i$ based on a dyadic grid. Let

$$N_{j,\ell} := \sum_{i=2}^{|\mathcal{A}_{m_\ell}|} \mathbf{1}\{\Delta_i \in [2^{-(j+1)} - \delta_0(\hat{a}_{\ell,m_\ell}), 2^{-j} - \delta_0(\hat{a}_{\ell,m_\ell}))\},$$

where $j$ ranges from 0 to

$$J := \left\lceil 1 \vee \frac{1}{\beta} \log \left( \frac{(t_{\ell+1}-t_\ell^{m_\ell})^{\frac{\beta}{\beta+1}}}{42^\beta \log^{3\beta}(T)} \right) \right\rceil$$

Next, the following lemma bounds $N_{j,\ell}$ in high probability using Freedman's inequality (Theorem 7).

**Lemma 12.** *(Proof in Subsection A.8) Let $\mathcal{E}_4$ be the event that the following hold:*

$$\forall j \in \{0, \ldots, J\}, \ell \in \{1, \ldots, \hat{L}-1\} : N_{j,\ell} \le \frac{3\kappa_2}{2} |\mathcal{A}_{m_\ell}| \cdot 2^{-j\beta} + \frac{13}{2} \log(2T^3) \tag{15}$$

$$\sum_{i=2}^{|\mathcal{A}_{m_\ell}|} \mathbf{1}\left\{ 4 \left( \frac{\log(T)}{t_{\ell+1}-t_\ell^{m_\ell}} \right)^{\frac{1}{\beta+1}} \le \Delta_i < 2^{-(J+1)} \right\} \le \frac{3\kappa_2}{2} |\mathcal{A}_{m_\ell}| \cdot 2^{-(J+1)\beta} + \frac{13}{2} \log(2T^3) \tag{16}$$

*Then, $\mathcal{E}_4$ occurs with probability at least $1 - 1/T$.*

The next lemma relates the cutoff $\Delta_i \ge 4\alpha$ to the quantity $2^{-(J+2)}$ showing that every gap $\Delta_i$ such that $\Delta_i \ge 4\alpha$ lies in one of the bins of the dyadic grid.

**Lemma 13.** *(Proof in Subsection A.9) We have*

$$2^{-(J+2)} - \delta_0(\hat{a}_{\ell,m_\ell}) > 4\alpha.$$

Now, Theorem 13 implies $2^{-(J+1)} - \delta_0(\hat{a}_{\ell,m_\ell}) > 4\alpha$. Thus, every gap $\Delta_i$ such that $\Delta_i/4 \ge \frac{\log^3(T)}{(t_{\ell+1}-t_\ell^{m_\ell})^{\frac{1}{\beta+1}}}$ lies in some interval $[2^{-(j+1)} - \delta_0(\hat{a}_{\ell,m_\ell}), 2^{-j} - \delta_0(\hat{a}_{\ell,m_\ell}))$ for some $j \in \{0, \ldots, J\}$ or lies in the interval $[4\alpha, 2^{-(J+1)} - \delta_0(\hat{a}_{\ell,m_\ell}))$. Thus, we bound

$$\sum_{i=2}^{|\mathcal{A}_{m_\ell}|} \frac{\log(T)}{\Delta_i} \cdot \mathbf{1}\left\{ \frac{\Delta_i}{4} \ge \alpha \right\} \le \sum_{j=0}^J N_{j,\ell} \cdot (2^{-(j+1)} - \delta_0(\hat{a}_{\ell,m_\ell}))^{-1} \cdot \log(T)$$

$$+ \log(T) \cdot \alpha^{-1} \sum_{i=2}^{|\mathcal{A}_{m_\ell}|} \mathbf{1}\{\alpha \le \Delta_i < 2^{-(J+1)} - \delta_0(\hat{a}_{\ell,m_\ell})\} \tag{17}$$

Now, by Theorem 13, we have for all $j \in \{0, \ldots, J\}$:

$$2^{-(j+1)} - \delta_0(\hat{a}_{\ell,m_\ell}) > 2^{-(j+2)}.$$

Combining the above with Theorem 12, we have (17) is order

$$\log(T) \cdot \alpha^{-1} \left( \kappa_2 |\mathcal{A}_{m_\ell}| \cdot 2^{-(J+1)\beta} + \log(T) \right) + \log(T) \sum_{j=0}^{J} \kappa_2 \cdot |\mathcal{A}_{m_\ell}| \cdot 2^{j(1-\beta)+2} + 2^{(j+1)} \log(T).$$

We have

$$2^{-(J+1)\beta} \le 2^{-\beta} \left( \frac{1}{2^\beta} \wedge \left( \frac{42^\beta \log^{3\beta}(T)}{(t_{\ell+1} - t_\ell^{m_\ell})^{\frac{\beta}{\beta+1}}} \right) \right) \le \frac{42^\beta \log^{3\beta}(T)}{2^\beta (t_{\ell+1} - t_\ell^{m_\ell})^{\frac{\beta}{\beta+1}}}.$$

Then,

$$\begin{aligned}
\kappa_2 \cdot \log(T) \cdot \alpha^{-1} \cdot 2^{-(J+1)\beta} &\le c_3 \kappa_2 \cdot (t_{\ell+1} - t_\ell^{m_\ell})^{\frac{1-\beta}{\beta+1}} \log^{3\beta-3}(T) \\
&\le c_4 \kappa_2 \log^{3(1-\beta)}(T) \cdot \log^{3\beta-3}(T) \\
&= c_5 \log(T),
\end{aligned}$$

where the second inequality follows from (8) of Theorem 6 and $\beta \ge 1$, and the third inequality follows from $\log(T) \ge \kappa_2$ by choosing $T$ sufficiently large. Thus,

$$\log(T) \cdot \alpha^{-1} \left( \kappa_2 |\mathcal{A}_{m_\ell}| \cdot 2^{-(J+1)\beta} + \log(T) \right) \le c_6 |\mathcal{A}_{m_\ell}| \log(T).$$

Next,

$$\begin{aligned}
\kappa_2 \log(T) \sum_{j=0}^{J} 2^{j(1-\beta)+2} &\le 4(J+1) \cdot 2^{J(1-\beta)} \log^2(T) \\
&\le 8 \log^2(T) \left( 2^{1-\beta} \vee \frac{(t_{\ell+1} - t_\ell^{m_\ell})^{\frac{1-\beta}{\beta+1}}}{42^{1-\beta} \log^{2(1-\beta)}(T)} \right) \left( 1 \vee \log \left( \frac{(t_{\ell+1} - t_\ell^{m_\ell})^{\frac{1}{\beta+1}}}{42^\beta \log^3(T)} \right) \right) \\
&\le 8 \log^2(T) \left( 1 + \log(T) \right) \\
&\le c_7 \log^3(T),
\end{aligned}$$

where the first inequality follows from $\kappa_2 \le \log(T)$, and the third inequality follows from $\beta \ge 1$ and (8) of Theorem 6.

Thus, (17) is at most order $|\mathcal{A}_{m_\ell}| \log^3(T)$. Thus, choosing $C_1$ large enough in Algorithm 1 of Algorithm 1, we have that (14) holds. Following earlier discussion, this means the subsampled bandit environment over the last block $[t_\ell^m, t_{\ell+1})$ is *not mildly corrupt* (Theorem 1). Then, for any $\mu(\cdot) : \mathcal{A}_{m_\ell} \to [0,1]$, there always exists $t' \in [t_\ell^m, t_{\ell+1})$ and $a' \in \mathcal{A}_{m_\ell}$ such that

$$|\mu_{t'}(a') - \mu(a')| > \frac{\log^3(T)}{(t_{\ell+1} - t_\ell^{m_\ell})^{\frac{1}{\beta+1}}}.$$

This means that since the adversary can only modify the rewards of arms $a_t$ after the round $t$, we have

$$\begin{aligned}
\sum_{t=t_\ell}^{t_{\ell+1}-1} |\mu_t(a_{t-1}) - \mu_{t-1}(a_{t-1})| &\ge \sum_{t=t_\ell^{m_\ell}}^{t_{\ell+1}-1} |\mu_t(a_{t-1}) - \mu_{t-1}(a_{t-1})| \\
&\ge \sum_{t=t_\ell^{m_\ell}}^{t_{\ell+1}-1} |\mu_t(a') - \mu_{t-1}(a')| \\
&\ge |\mu_{t'}(a') - \mu_0(a')| \\
&\ge \frac{\log^3(T)}{(t_{\ell+1} - t_\ell^{m_\ell})^{\frac{1}{\beta+1}}} \\
&\ge \frac{\log^3(T)}{(t_{\ell+1} - t_\ell)^{\frac{1}{\beta+1}}}.
\end{aligned}$$

□

## A.5. Relating Episodes to Non-Stationarity Measures

Now, we show how to derive

$$\sum_{\ell=1}^{\hat{L}} (t_{\ell+1} - t_\ell)^{\frac{\beta}{\beta+1}} \log^3(T) \leq \left( (L+1)^{\frac{1}{\beta+1}} \cdot T^{\frac{\beta}{\beta+1}} \wedge (V^{\frac{1}{\beta+2}} T^{\frac{\beta+1}{\beta+2}} + T^{\frac{\beta}{\beta+1}}) \right) \cdot \log^3(T),$$

from the fact that, as we have shown by Theorem 10, for all episodes $\ell \in [\hat{L} - 1]$

$$\sum_{t=t_\ell}^{t_{\ell+1}-1} |\mu_t(a_{t-1}) - \mu_{t-1}(a_{t-1})| \geq \frac{\log^3(T)}{(t_{\ell+1} - t_\ell)^{\frac{1}{\beta+1}}}.$$

We first show the bound in terms of total variation $V$. Let $V_{[s_1, s_2)} := \sum_{t=s_1}^{s_2-1} |\mu_t(a_{t-1}) - \mu_{t-1}(a_{t-1})|$. Then, by using Hölder's inequality:

$$\sum_{\ell=1}^{\hat{L}} (t_{\ell+1} - t_\ell)^{\frac{\beta}{\beta+1}} \log^3(T) \leq T^{\frac{\beta}{\beta+1}} \log^3(T) + \left( \sum_{\ell=1}^{\hat{L}-1} \frac{\log^3(T)}{(t_{\ell+1} - t_\ell)^{\frac{1}{\beta+1}}} \right)^{\frac{1}{\beta+2}} \left( \sum_{\ell=1}^{\hat{L}-1} (t_{\ell+1} - t_\ell) \log^{3-\frac{3}{\beta+2}}(T) \right)^{\frac{\beta+1}{\beta+2}}$$

$$\leq T^{\frac{\beta}{\beta+1}} \log^3(T) + \left( \sum_{\ell=1}^{\hat{L}-1} V_{[t_\ell, t_{\ell+1})} \right)^{\frac{1}{\beta+2}} T^{\frac{\beta+1}{\beta+2}} \cdot \log^{\frac{3(\beta+1)^2}{(\beta+2)^2}}(T)$$

$$\leq T^{\frac{\beta}{\beta+1}} \log^3(T) + V^{\frac{1}{\beta+2}} T^{\frac{\beta+1}{\beta+2}} \cdot \log^{\frac{3(\beta+1)^2}{(\beta+2)^2}}(T)$$

To show the bound in terms of $L$, we use Jensen's inequality on the function $x \mapsto x^{\frac{\beta}{\beta+1}}$ combined with the fact that the number of episodes $\hat{L} \leq L + 1$ by virtue of Theorem 10.

## A.6. A Sketch of Modifications Required for Proving Theorem 2 for $\beta < 1$

Next, we describe how to show a suboptimal regret bound of order $\sum_{\ell=1}^{\hat{L}} \sqrt{t_{\ell+1} - t_\ell}$ in the setting of $\beta < 1$. From here, using the steps of Subsection A.5 with $\beta = 1$, it is straightforward to show a regret bound of $\sqrt{(L+1)T} \wedge (V^{1/3} T^{2/3} + \sqrt{T})$. For the sake of redundancy, we only give here a sketch of the modifications to the argument required.

We'll first describe at a high level the difficulty of showing a $\sum_{\ell=1}^{\hat{L}} (t_{\ell+1} - t_\ell)^{\frac{\beta}{\beta+1}}$ regret bound using the same arguments of the previous sections for $\beta < 1$. In fact, the only place where $\beta \geq 1$ was used is in the final step where we bound (17). In particular, when bounding the sum $\sum_{j=0}^{J} 2^{j(1-\beta)+2} \leq 4(J+1) \cdot 2^{J(1-\beta)}$, for $\beta \geq 1$, $2^{J(1-\beta)}$ is a constant. However, for $\beta < 1$, we may incur an extra $(t_{\ell+1} - t_\ell^{m_\ell})^{\frac{1-\beta}{\beta+1}}$ term in the final regret bound due to this term. Thus, this argument would only yield a suboptimal regret bound. Interestingly, Bayati et al. (2020, cf. E.1.2) and Kim et al. (2024) also face this difficulty in bounding similar terms which leads to a suboptimal rate.

However, we can still attain a $\sqrt{t_{\ell+1} - t_\ell}$ per-episode regret bound using an altered subsampling rate $S_m := \lceil 2^{m \cdot \beta/2} \rceil \wedge 2^m$ which effectively "targets" a $\sqrt{t_{\ell+1} - t_\ell}$ regret rate.

Going into more detail, a first key fact, as observed in Bayati et al. (2020), is that subsampling $S_m$ arms ensures an arm with gap $\tilde{O}((t_{\ell+1} - t_\ell)^{-1/2})$. This can be seen analogously to the proof of Theorem 11 in Subsection A.7 where letting $\beta = 1$ and $|\mathcal{A}_{m_\ell}| \gtrsim 2^{m_\ell \cdot \beta/2}$ in Subsection A.7 establishes that the best initial arm has gap at most $\log^3(T) \cdot 2^{-m_\ell \cdot \beta/2}$.

Then, the key fact to show will be an analogue of Theorem 10: with probability at least $1 - 4/T$, for all $\ell \in [\hat{L} - 1]$:

$$\sum_{t=t_\ell}^{t_{\ell+1}-1} |\mu_t(a_{t-1}) - \mu_{t-1}(a_{t-1})| \geq \frac{\log^3(T)}{\sqrt{t_{\ell+1} - t_\ell}}. \tag{18}$$

To show this, we'll essentially repeat the arguments of Subsection A.4 except specializing "$\beta = 1$" to target a bound scaling like $\sqrt{t_{\ell+1} - t_\ell}$. Note that using the scaling $|\mathcal{A}_m| \vee 2^{m/2}$ in the regret threshold for the changepoint detection test (Algorithm 1 of Algorithm 1) is crucial here as $|\mathcal{A}_m| \propto 2^{m \cdot \beta/2} \ll 2^{m/2}$ in the case of $\beta < 1$.

First, if the dynamic regret of the best initial arm $\hat{a}_{\ell, m_\ell}$ is larger than $\frac{C_1}{4} 2^{m_\ell/2} \cdot \log^3(T)$, then using the previously established fact that $\delta_0(\hat{a}_{\ell, m_\ell}) \leq \log^3(T) \cdot 2^{-m_\ell \cdot \beta/2}$, we have that (18) holds.

Next, following the argument structure of Subsection A.4, suppose $\sum_{t=t_\ell^{m_\ell}}^{t_{\ell+1}-1} \delta_t(\hat{a}_{\ell, m_\ell}) \leq \frac{C_1}{4} 2^{m_\ell/2} \log^3(T)$ but $\sum_{t=t_\ell^{m_\ell}}^{t_{\ell+1}-1} \mu_t(\hat{a}_{\ell, m_\ell}) - \mu_t(a_t) \geq \frac{C_1}{4} 2^{m_\ell/2} \log^3(T)$. Then, we invoke Assumption 2 with $\alpha = \log^3(T) \cdot (t_{\ell+1} - t_\ell^{m_\ell})^{-1/2}$ and use the same dyadic gridding argument with

$$J := \left\lceil 1 \vee \log\left(\frac{(t_{\ell+1} - t_\ell^{m_\ell})^{1/2}}{42 \log^3(T)}\right)\right\rceil.$$

Then, observe the bounds

$$\alpha^{-1} \cdot 2^{-(J+1)\beta} \lesssim \frac{(t_{\ell+1} - t_\ell^{m_\ell})^{1/2}}{\log^3(T)} \cdot \frac{\log^3(T)}{(t_{\ell+1} - t_\ell^{m_\ell})^{1/2}} \lesssim O(1)$$

$$J \cdot 2^{J(1-\beta)} \cdot |\mathcal{A}_{m_\ell}| \lesssim 2^{m_\ell \cdot \beta/2} \cdot (t_{\ell+1} - t_\ell^{m_\ell})^{(1-\beta)/2} \leq 2^{m_\ell/2}.$$

Thus, we can show $\sum_{t=t_\ell^{m_\ell}}^{t_{\ell+1}} \mu_t(\hat{a}_{\ell, m_\ell}) - \mu_t(a_t) < \frac{C_1}{4} 2^{m_\ell/2} \log^3(T)$ if (18) does not hold, which contradicts our earlier supposition. This means (18) holds.

## A.7. Proof of Theorem 11

Let $\mu_0(a)$ denote the initial mean reward of arm $a$. By Assumption 1, we have

$$\mathbb{P}\left(\mu_0(\hat{a}_{\ell, m_\ell}) \leq 1 - \frac{\log^3(T)}{2^{m_\ell \cdot \frac{1}{\beta+1}}}\right) = \mathbb{P}\left(\max_{a \in \mathcal{A}_{m_\ell}} \mu_0(a) \leq 1 - \frac{\log^3(T)}{2^{m_\ell \cdot \frac{1}{\beta+1}}}\right)$$

$$\leq \left(1 - \mathbb{P}\left(\mu_0(a) > 1 - \frac{\log^3(T)}{2^{m_\ell \cdot \frac{1}{\beta+1}}}\right)\right)^{|\mathcal{A}_{m_\ell}|}$$

$$\leq \left(1 - \kappa_1 \cdot \frac{\log^{3\beta}(T)}{2^{m_\ell \cdot \frac{\beta}{\beta+1}}}\right)^{|\mathcal{A}_{m_\ell}|}$$

$$\leq \exp\left(-|\mathcal{A}_{m_\ell}| \cdot \frac{\log(T)}{2^{m_\ell \cdot \frac{\beta}{\beta+1}}}\right)$$

$$\leq \frac{1}{T^2},$$

where the fourth line follows from assuming WLOG that $\log(T) \geq \kappa_1^{-\frac{1}{2\beta}}$ and $\beta \geq 1$, and the last inequality from $|\mathcal{A}_{m_\ell}| \geq 2^{1 + m_\ell \cdot \frac{\beta}{\beta+1}}$.

## A.8. Proof of Theorem 12

By Freedman's inequality (Theorem 7), we have with probability at least $1/T^3$:

$$|N_{j,\ell} - \mathbb{E}[N_{j,\ell}]| \leq 3\sqrt{\mathbb{E}[N_{j,\ell}] \cdot \log(2T^3)} + 2\log(2T^3)$$

Using AM-GM, this yields,

$$N_{j,\ell} \leq \frac{3\mathbb{E}[N_{j,\ell}]}{2} + \frac{13}{2}\log(2T^3).$$

Finally,

$$\mathbb{P}(2^{-(j+1)} - \delta_0(\hat{a}_{\ell, m_\ell}) \leq \Delta_i < 2^{-j} - \delta_0(\hat{a}_{\ell, m_\ell})) = \mathbb{P}(2^{-(j+1)} \leq \delta_0(i) < 2^{-j}) \leq \mathbb{P}(1 - 2^{-j} < \mu_0(i)) \leq \kappa_2 \cdot 2^{-j\beta}.$$

Thus,

$$\mathbb{E}[N_{j,\ell}] \leq (|\mathcal{A}_{m_\ell}| - 1) \cdot \kappa_2 \cdot 2^{-j\beta}.$$

This shows (15). (16) is showed in a nearly identical manner.

### A.9. Proof of Theorem 13

Recall $\alpha := \frac{\log^3(T)}{(t_{\ell+1} - t_\ell^{m_\ell})^{\frac{1}{\beta+1}}}$ and let $\delta_0 := \delta_0(\hat{a}_{\ell,m_\ell})$ to ease notation. First, suppose $J = 1$. Then, by Theorem 11

$$2^{-3} - \delta_0 > 4\alpha \impliedby \frac{1}{8} > \frac{\log^3(T)}{2^{m_\ell \cdot \frac{1}{\beta+1}}} + \frac{4\log^3(T)}{(t_{\ell+1} - t_\ell^{m_\ell})^{\frac{1}{\beta+1}}}.$$

Now, since $\beta \geq 1$ and $2^{m_\ell} \geq t_{\ell+1} - t_\ell^{m_\ell}$, it suffices to show

$$\frac{1}{8} > \frac{5\log^3(T)}{(t_{\ell+1} - t_\ell^{m_\ell})^{\frac{1}{\beta+1}}} \iff (t_{\ell+1} - t_\ell^{m_\ell})^{\frac{1}{\beta+1}} > 40\log^3(T).$$

However, this last inequality is true from (8) of Theorem 6 for sufficiently large $C_1$.

Now, suppose $J > 1$. Rearranging the desired inequality we have

$$(\delta_0 + 4\alpha)^{-1} > 2^{J+2} \impliedby \frac{1}{4(\delta_0 + 4\alpha)} > 4 \vee 2 \cdot \frac{(t_{\ell+1} - t_\ell^{m_\ell})^{\frac{1}{\beta+1}}}{42\log^3(T)}$$

From (8) of Theorem 6, we have for sufficiently large $C_1$:

$$2 \cdot \frac{(t_{\ell+1} - t_\ell^{m_\ell})^{\frac{1}{\beta+1}}}{42\log^3(T)} > 4.$$

Thus, in light of Theorem 11 and the definition of $\alpha$ we want to show

$$\frac{21\log^3(T)}{(t_{\ell+1} - t_\ell^{m_\ell})^{\frac{1}{\beta+1}}} > \frac{4\log^3(T)}{(t_{\ell+1} - t_\ell^{m_\ell})^{\frac{1}{\beta+1}}} + \frac{16\log^3(T)}{(t_{\ell+1} - t_\ell^{m_\ell})^{\frac{1}{\beta+1}}},$$

which is always true.

## B. Elimination Algorithm Regret Analysis (Proofs of Theorem 4 and Theorem 5)

Following the notation of Section C, we let $c_0, c_1, c_2, \ldots$ denote constants not depending on $T$, $\kappa_1$, or $\kappa_2$ (Assumption 1).

### B.1. Details for the Proof of Theorem 4

As in the proof of Theorem 2 in Section A, we assume WLOG that $\log(T) \geq \kappa_1^{-(1 \wedge 2\beta)}$ or else we can bound the regret by a constant only depending on $\kappa_1$.

Following the proof outline of Subsection 5.3, we have by Freedman's inequality (Theorem 7) with probability at least $1 - 1/T$:

$$\sum_{t=t_\ell^m}^{t_\ell^{m+1}-1} \delta_t(a_t) \leq \sum_{t=t_\ell^m}^{t_\ell^{m+1}-1} \sum_{a \in \mathcal{G}_t} \frac{\delta_t(a)}{|\mathcal{G}_t|} + c_8 \cdot \left( \log(T) + \sqrt{\log(T) \sum_{t=t_\ell^m}^{t_\ell^{m+1}-1} \mathbb{E}[\delta_t(a_t) \mid \mathcal{H}_{t-1}]} \right)$$

$$\leq \frac{3}{2} \sum_{t=1}^{t_\ell^{m+1}-1} \sum_{a \in \mathcal{G}_t} \frac{\delta_t(a)}{|\mathcal{G}_t|} + c_9 \log(T),$$

where the second inequality uses AM-GM. Now, recalling that $t^a$ is the last round in block $[t_\ell^m, t_\ell^{m+1})$ that arm $a \in [K]$ is retained, we have:

$$\sum_{t=t_\ell^m}^{t^a} \frac{\delta_t(a)}{|\mathcal{G}_t|} \le \left( \max_{s \in [t_\ell^m, t^a]} \frac{1}{|\mathcal{G}_s|} \right) \sum_{t=t_\ell^m}^{t^a} \delta_t(a). \tag{19}$$

We next again use Freedman's inequality (Theorem 7) to relate $\sum_{t \in I} \delta_t(a)$ to $\sum_{t \in I} \hat{\delta}_t^{\mathrm{IW}}(a)$. For the variance bound, we have

$$\begin{aligned}
\mathbb{E}[(\delta_t(a) - \hat{\delta}_t^{\mathrm{IW}}(a))^2 | \mathcal{H}_{t-1}] &\le \mathbb{E}[(\hat{\delta}_t^{\mathrm{IW}})^2(a) | \mathcal{H}_{t-1}] \\
&= \mathbb{E}[|\mathcal{G}_t|^2 \cdot (1 - Y_t(a_t))^2 \cdot \mathbf{1}\{a_t = a\} | \mathcal{H}_{t-1}] \\
&\le |\mathcal{G}_t|^2 \cdot \mathbb{E}[(1 - Y_t(a)) | \mathcal{H}_{t-1}] \cdot \mathbb{E}[\mathbf{1}\{a_t = a\} | \mathcal{H}_{t-1}] \\
&= |\mathcal{G}_t|^2 \cdot \delta_t(a) \cdot \frac{1}{|\mathcal{G}_t|} \\
&= \delta_t(a) \cdot |\mathcal{G}_t|.
\end{aligned}$$

We also have $\max_{s \in I} |\delta_s(a) - \hat{\delta}_s^{\mathrm{IW}}(a)| \le \max_{s \in I} |\mathcal{G}_s|$ and $\mathbb{E}[\hat{\delta}_t^{\mathrm{IW}}(a) | \mathcal{H}_{t-1}] = \delta_t(a)$.

From the above and Freedman's inequality, we can show that w.p. at least $1 - 1/T$, for any interval $I \subseteq [T]$ on which arm $a$ is retained:

$$\begin{aligned}
\left| \sum_{t=t_\ell^m}^{t^a} \delta_t(a) - \hat{\delta}_t^{\mathrm{IW}}(a) \right| &\le c_{10} \left( \max_{t \in [t_\ell^m, t^a]} |\mathcal{G}_t| \log(T) + \sqrt{\log(T) \sum_{t=t_\ell^m}^{t^a} \delta_t(a) \cdot |\mathcal{G}_t|} \right) \\
&\le \frac{1}{2} \sum_{t=t_\ell^m}^{t^a} \delta_t(a) + c_{11} \max_{t \in [t_\ell^m, t^a]} |\mathcal{G}_t| \log(T), \tag{20}
\end{aligned}$$

where again we use AM-GM in the second inequality. Moving the $\frac{1}{2} \sum_t \delta_t(a)$ to the other side, we get

$$\sum_{t=t_\ell^m}^{t^a} \delta_t(a) \le 2 \sum_{t=t_\ell^m}^{t^a} \hat{\delta}_t^{\mathrm{IW}}(a) + c_{12} \log(T) \max_{t \in [t_\ell^m, t^a]} |\mathcal{G}_t|.$$

Plugging the above into (19), we have

$$\begin{aligned}
\sum_{t=t_\ell^m}^{t^a} \frac{\delta_t(a)}{|\mathcal{G}_t|} &\le \left( \max_{s \in [t_\ell^m, t^a]} \frac{c_{13}}{|\mathcal{G}_s|} \right) \left( \sum_{t=t_\ell^m}^{t^a} \hat{\delta}_t^{\mathrm{IW}}(a) + \log(T) \max_{t \in [t_\ell^m, t^a]} |\mathcal{G}_t| \right) \\
&\le c_{14} \max_{s \in [t_\ell^m, t^a]} \frac{|\mathcal{A}_m|}{|\mathcal{G}_s|} \log(T),
\end{aligned}$$

where the second inequality is from the elimination guarantee (Algorithm 2 of Algorithm 2) and $\max_{t \in [t_\ell^m, t^a]} |\mathcal{G}_t| = |\mathcal{A}_m|$.

We next have

$$\begin{aligned}
\sum_{a \in \mathcal{A}_m} \max_{s \in [t_\ell^m, t^a - 1]} \frac{1}{|\mathcal{G}_s|} &= \sum_{a \in \mathcal{A}_m} \frac{1}{|\mathcal{G}_{t^a-1}|} \cdot \mathbf{1}\{t^a < t_\ell^{m+1} - 1\} + \frac{1}{|\mathcal{G}_{t_\ell^{m+1}-1}|} \cdot \mathbf{1}\{t^a = t_\ell^{m+1} - 1\} \\
&\le \sum_{i=1}^{|\mathcal{A}_m|} \frac{1}{i} + \frac{|\mathcal{G}_{t_\ell^{m+1}-1}|}{|\mathcal{G}_{t_\ell^{m+1}-1}|} \\
&\le 1 + \log(|\mathcal{A}_m|),
\end{aligned}$$

Now, combining the above with our previous bound and summing over arms $a$, we have:

$$\sum_{a \in \mathcal{A}_m} \sum_{t=t_\ell^m}^{t_\ell^{m+1}-1} \frac{\delta_t(a)}{|\mathcal{G}_t|} \cdot \mathbf{1}\{a \in \mathcal{G}_t\} \le c_{15} |\mathcal{A}_m| \cdot \log^2(T).$$

We know that for any $m \neq m_\ell$, we have $|\mathcal{A}_m| = \left\lceil 2^{(m+1) \cdot \frac{\beta}{\beta+1}} \cdot \log(T) \right\rceil$. We also have

$$|\mathcal{A}_{m_\ell}| \leq 2|\mathcal{A}_{m_\ell-1}| = 2 \left\lceil (2(t_\ell^{m_\ell+1} - t_\ell^{m_\ell}))^{\frac{\beta}{\beta+1}} \log(T) \right\rceil.$$

Then, summing $|\mathcal{A}_m| \log^2(T)$ over $m \in [m_\ell]$ and $\ell \in [\hat{L}]$ gives us the regret bound of order $\sum_{\ell=1}^{\hat{L}} (t_{\ell+1} - t_\ell)^{\frac{\beta}{\beta+1}} \log^3(T)$.

## B.2. Details for the Proof of Theorem 5

We define event $\mathcal{E}_5$ such that, for any $I \subseteq [T]$,

$$\left| \sum_{t \in I} \delta_t(a) - \hat{\delta}_t^{\mathrm{IW}}(a) \right| \leq \frac{1}{2} \sum_{t \in I} \delta_t(a) + c_{16} \log(T) \max_{t \in I} |\mathcal{G}_t|, \tag{21}$$

which holds with probability at least $1 - 1/T$, by the same reasoning as (20) We consider episode $\ell \in [\hat{L}]$ that terminates in a restart and let $m_\ell$ be the index of the last block in $\ell$-th episode and $t_\ell$ be the start time of the $\ell$-th episode.

● **Bounding the Number of Episodes.** We claim the number of episodes is at most the number of significant shifts or $\hat{L} \leq \tilde{L} + 1$. In particular, we prove that a significant shift must have occurred over some block in each episode concluding with a restart. Let $t^a$ be the time when arm $a$ is eliminated. For the $m_\ell$-th block $[t_\ell^{m_\ell}, t_{\ell+1})$, on $\mathcal{E}_5$, for each arm $a \in \mathcal{A}_{m_\ell}$ we have by (21) and Algorithm 2 of Algorithm 2 with $C_2 > 0$ large enough::

$$\sum_{s=t_\ell^{m_\ell}}^{t^a} \delta_t(a) \geq \frac{2}{3} \left( \sum_{s=t_\ell^{m_\ell}}^{t^a} \hat{\delta}^{\mathrm{IW}}(a) - c_{18} \log(T) \max_{t \in [t_\ell^{m_\ell}, t^a]} |\mathcal{G}_t| \right) \geq c_{17} |\mathcal{A}_{m_\ell}| \log(T).$$

Then, for all $a \in \mathcal{A}_{m_\ell}$ where $|\mathcal{A}_{m_\ell}| = \left\lceil 2^{(m+1) \cdot \frac{\beta}{\beta+1}} \log(T) \right\rceil \geq (t_{\ell+1} - t_\ell^{m_\ell})^{\frac{\beta}{\beta+1}} \log(T) \geq (t^a - t_\ell^{m_\ell})^{\frac{\beta}{\beta+1}} \log(T)$, we have

$$\sum_{s=t_\ell^{m_\ell}}^{t^a} \delta_t(a) > 3(t_{\ell+1} - t_\ell^{m_\ell})^{\frac{\beta}{\beta+1}} \log^3(T) \geq 3(t^a - t_\ell^{m_\ell})^{\frac{\beta}{\beta+1}} \log^3(T), \tag{22}$$

which implies that (5) is violated for $\log(T) \geq \kappa_1^{-1}$, meaning arm $a$ is unsafe. By Theorem 3, a significant shift must have occurred within the block $[t_\ell^{m_\ell}, t_{\ell+1})$.

Now, by considering that the $\hat{L}$-th episode may end by reaching the horizon $T$ rather than restarting, which does not ensure a significant shift, we conclude that $\hat{L} \leq \tilde{L} + 1$. Therefore, by using Jensen's inequality, we have w.p. at least $1 - T^{-1}$,

$$\sum_{\ell=1}^{\hat{L}} (t_{\ell+1} - t_\ell)^{\frac{\beta}{\beta+1}} \leq \hat{L}^{\frac{1}{\beta+1}} T^{\frac{\beta}{\beta+1}} \leq (\tilde{L}+1)^{\frac{1}{\beta+1}} T^{\frac{\beta}{\beta+1}}. \tag{23}$$

● **Bounding the Per-Episode Variation.** Next, we show that, on event $\mathcal{E}_5$ where (21) holds, the total rotting variation over episode $[t_\ell, t_{\ell+1})$ is at least

$$\sum_{t=t_\ell}^{t_{\ell+1}-1} (\mu_{t-1}(a_{t-1}) - \mu_t(a_{t-1}))_+ \geq \frac{\log^3(T)}{(t_{\ell+1} - t_\ell)^{\frac{1}{\beta+1}}}.$$

From (22), we have for all $a \in \mathcal{A}_{m_\ell}$, there exists $t' \in [t_\ell^{m_\ell}, t^a]$ such that

$$\delta_{t'}(a) \geq \frac{3 \log^3(T)}{(t^a - t_\ell^{m_\ell})^{\frac{1}{\beta+1}}} \geq \frac{3 \log^3(T)}{(t_{\ell+1} - t_\ell^{m_\ell})^{\frac{1}{\beta+1}}}. \tag{24}$$

Recall $\mu_0(a)$ is the initial mean reward of arm $a$. Let $\hat{a}_{\ell,m}$ be the best arm in terms of initial reward value $\mu_0(\cdot)$ among the arms $\mathcal{A}_m$ subsampled for block $[t_\ell^m, t_\ell^{m+1})$. We have

$$\mathbb{P}\left(\mu_0(\hat{a}_{\ell,m_\ell}) < 1 - \frac{2\log^3(T)}{2^{m_\ell \cdot \frac{1}{\beta+1}}}\right) = \mathbb{P}\left(\max_{a \in \mathcal{A}_{m_\ell}} \mu_0(a) < 1 - \frac{2\log^3(T)}{2^{m_\ell \cdot \frac{1}{\beta+1}}}\right)$$

$$\leq \left(1 - \kappa_1 \frac{2\log^{3\beta}(T)}{2^{m_\ell \cdot \frac{\beta}{\beta+1}}}\right)^{|\mathcal{A}_{m_\ell}|}$$

$$\leq \exp\left(-|\mathcal{A}_{m_\ell}| \cdot \frac{2\kappa_1 \log^{3\beta}(T)}{2^{m_\ell \cdot \frac{\beta}{\beta+1}}}\right)$$

$$\leq \frac{1}{T^2},$$

where the last inequality follows from $\log(T) \geq \kappa_1^{-\frac{1}{2\beta}}$ and $|\mathcal{A}_{m_\ell}| \geq 2^{m_\ell \cdot \frac{\beta}{\beta+1}} \cdot \log(T)$. Then, we define $\mathcal{E}_6$ to be the corresponding high-probability event

$$\{\forall \ell \in [\hat{L}] : 1 - \mu_0(\hat{a}_{\ell,m_\ell}) \leq 2\log^3(T) \cdot 2^{-m_\ell \cdot \frac{1}{\beta+1}}\},$$

which holds with at least probability of $1 - 1/T$. On $\mathcal{E}_6$, we have

$$\mu_0(\hat{a}_{\ell,m_\ell}) \geq 1 - \frac{2\log^3(T)}{2^{m_\ell \cdot \frac{1}{\beta+1}}} \geq 1 - \frac{2\log^3(T)}{(t_{\ell+1} - t_\ell^{m_\ell})^{\frac{1}{\beta+1}}}. \tag{25}$$

From (24) and (25), we can observe that the mean reward of arm $\hat{a}_{\ell,m_\ell}$ must have moved by amount at least

$$\frac{\log^3(T)}{(t_{\ell+1} - t_\ell^{m_\ell})^{\frac{1}{\beta+1}}} \geq \frac{\log^3(T)}{(t_{\ell+1} - t_\ell)^{\frac{1}{\beta+1}}},$$

over the course of the block, implying

$$\mu_0(\hat{a}_{\ell,m_\ell}) - \mu_{t'}(\hat{a}_{\ell,m_\ell}) \geq \frac{\log^3(T)}{(t_{\ell+1} - t_\ell^{m_\ell})^{\frac{1}{\beta+1}}} \geq \frac{\log^3(T)}{(t_{\ell+1} - t_\ell)^{\frac{1}{\beta+1}}}.$$

Since the adversary can only modify the rewards of an arm $a_t$ after the round $t$ it is played, the movement must have occurred on rounds where $\hat{a}_{\ell,m_\ell}$ was chosen by the agent. Thus,

$$\sum_{t=t_\ell}^{t_{\ell+1}-1} (\mu_{t-1}(a_{t-1}) - \mu_t(a_{t-1}))_+ \geq \sum_{t=t_\ell}^{t_{\ell+1}-1} (\mu_{t-1}(\hat{a}_{\ell,m_\ell}) - \mu_t(\hat{a}_{\ell,m_\ell}))_+ \geq \mu_0(\hat{a}_{\ell,m_\ell}) - \mu_{t'}(\hat{a}_{\ell,m_\ell}) \geq \frac{\log^3(T)}{(t_{\ell+1} - t_\ell)^{\frac{1}{\beta+1}}}.$$

Next, this lower bound on the per-episode variation gives us, in an identical manner Subsection A.5, a cumulative regret bound:

$$\sum_{\ell=1}^{\hat{L}} (t_{\ell+1} - t_\ell)^{\frac{\beta}{\beta+1}} \log^3(T) \leq \log^3(T) \cdot \left(V_R^{\frac{1}{\beta+2}} T^{\frac{\beta+1}{\beta+2}} + T^{\frac{\beta}{\beta+1}}\right).$$

# C. Verifying Assumption 2 for UCB

Here, we show the UCB algorithm satisfies Assumption 2. The proof will mostly follow the standard proof for showing the classsical logarithmic regret bound (e.g. Lattimore and Szepesvári, 2020, Section 7.1), with some small modifications to account for the mild corruption (Theorem 1).

We first present a variant of the UCB1 Algorithm of Auer et al. (2002).

**Theorem 14.** *Algorithm 3 satisfies Assumption 2.*

---

**Algorithm 3:** Variant of UCB1 (Algoritm 3 of Lattimore and Szepesvári (2020))

---

1 **Input**: number of arms $K$, horizon $T$.

2 **for** $t = 1, \ldots, T$ **do**

3      Let $N_{t-1}(a) := \sum_{s=1}^{t-1} \mathbf{1}\{a_s = a\}$.

4      Let $\mathsf{UCB}_a(t-1) := \begin{cases} \infty & N_{t-1}(a) = 0 \\ \frac{1}{N_{t-1}(a)} \sum_{s=1}^{t-1} \mathbf{1}\{a_s = a\} \cdot Y_s(a) + \sqrt{\frac{2 \log(T)}{N_{t-1}(a)}} & \text{otherwise} \end{cases}$.

5      Play arm $a_t = \arg\max_{a \in [K]} \mathsf{UCB}_a(t-1)$.

---

*Proof.* Our goal is to establish a high-probability regret bound over $t \leq T$ rounds. First, using Theorem 1, we may write the regret as

$$\max_{a \in \mathcal{A}_0} \sum_{s=1}^{t} \mu_s(a) - \mu_s(a_s) \leq \max_{a \in \mathcal{A}_0} \sum_{s=1}^{t} \mu(a) - \mu(a_s) + 2t \cdot \alpha.$$

Recall $N_t(a) := \sum_{s=1}^{t} \mathbf{1}\{a_s = a\}$. Let $\Delta_a$ denote the gap of arm $a \in \mathcal{A}_0$. Then, we can decompose the regret based on whether the gap of each arm is large or small and write:

$$\max_{a \in \mathcal{A}_0} \sum_{s=1}^{t} \mu(a) - \mu(a_s) \leq \sum_{a \in \mathcal{A}_0} \Delta_a \cdot N_t(a) \cdot \mathbf{1}\{\Delta_a > 4\alpha\} + 4t \cdot \alpha. \tag{26}$$

It then remains to bound $N_t(a)$ w.h.p for each $a \in \mathcal{A}_0$ such that $\Delta_a > \alpha$. WLOG, suppose arm $a = 1$ is the best arm among the arms in $\mathcal{A}_0$, which we'll index by the set $[|\mathcal{A}_0|]$.

We claim with probability at least $1 - 1/T$, for $t$ such that $N_t(a) > \left\lceil \frac{32 \log(T)}{\Delta_a^2} \right\rceil$:

$$\mathsf{UCB}_a(t) < \mathsf{UCB}_1(t). \tag{27}$$

This will allow us to conclude by bounding $N_t(a) \leq \left\lceil \frac{32 \log(T)}{\Delta_a^2} \right\rceil$ in (26).

Let $\hat{\mu}_t(a) := \sum_{s=1}^{t} Y_s(a) \cdot \mathbf{1}\{a_s = a\}$ and let $\overline{\mu}_t(a) := \sum_{s=1}^{t} \mu_s(a) \cdot \mathbf{1}\{a_s = a\}$. Then, by Azuma-Hoeffding inequality and a union bound we have

$$\mathbb{P}\left( \forall s \in [t] : |\hat{\mu}_s(a) - \overline{\mu}_s(a)| \geq \sqrt{\frac{N_s(a) \cdot \log(2T^2)}{2}} \right) \leq 1/T.$$

Going forward, suppose the above concentration bound holds.

Now, to show the claim, we first note for $t$ such that $N_t(a) > \left\lceil \frac{32 \log(T)}{\Delta_a^2} \right\rceil$ and $a \neq 1$ such that $\Delta_a > 4\alpha$:

$$\begin{aligned} \mathsf{UCB}_a(t) &= \hat{\mu}_t(a) + \sqrt{\frac{2 \log(T)}{N_t(a)}} \\ &\leq \overline{\mu}_t(a) + \sqrt{\frac{2 \log(T)}{N_t(a)}} + \sqrt{\frac{\log(2T^2)}{2N_t(a)}} \\ &\leq \mu(a) + \sqrt{\frac{2 \log(T)}{N_t(a)}} + \sqrt{\frac{\log(2T^2)}{2N_t(a)}} + \alpha \\ &< \mu(a) + \frac{\Delta_a}{4} + \frac{\Delta_a}{4} + \frac{\Delta_a}{4} \\ &= \mu(a) + \frac{3\Delta_a}{4}. \end{aligned}$$

where the third inequality is by Theorem 1. Meanwhile,

$$\begin{aligned}
\mathsf{UCB}_1(t) &= \hat{\mu}_t(1) + \sqrt{\frac{2\log(T)}{N_t(a)}} \\
&\geq \overline{\mu}_t(1) + \sqrt{\frac{2\log(T)}{N_t(a)}} - \sqrt{\frac{\log(2T^2)}{2N_t(a)}} \\
&> \mu(1) - \alpha \\
&\geq \mu(1) - \Delta_a/4.
\end{aligned}$$

Thus, claim (27) is shown. □

**Remark 6.** *Although not shown here for the sake of redundancy, a very similar regret analysis as Theorem 14 shows the Successive Elimination algorithm (Even-Dar et al., 2006) also satisfies Assumption 2.*

