# OpenReview forum: "Tracking Most Significant Shifts in Infinite-Armed Bandits"
_ICML.cc/2025/Conference — ICML 2025 poster_

### Official Review · Reviewer_T9xW · 2025-03-10

**Overall Recommendation:** 2

**Summary:**

The paper studies the non-stationary infinite-armed bandit problem where arms' mean rewards are initially sampled from a $\beta$-regular reservoir distribution and evolve under adversarial/non-stationary dynamics. Prior works focused on stationary rewards or specific non-stationary cases requiring prior knowledge of non-stationarity parameters. This work addresses general adversarial non-stationarity without parameter knowledge, relaxing distributional assumptions of the reservoir. The model captures scenarios with massive action spaces where rewards change adaptively based on played arms.

Main Algorithmic Ideas: The proposed framework introduces two key innovations:
1. Blackbox Reduction: Converts finite-armed MAB algorithms into parameter-free algorithms for infinite arms via dynamic subsampling. Uses empirical regret tracking to detect non-stationarity and trigger restarts.
2. Randomized Elimination: Enhances adaptation through a restarting elimination algorithm that detects "significant shifts" — intervals where all subsampled arms become suboptimal due to non-stationarity.

Main Results: Achieve the first optimal and parameter-free regret bounds for infinite-armed settings which are validated by the experiments.

**Claims And Evidence:**

The claims presented in this paper are all supported by corresponding theoretical proofs.

**Essential References Not Discussed:**

I believe the discussion regarding [Wei and Luo, 2021] in this work is significantly lacking. The black-box algorithm proposed in this work largely draws inspiration from the MASTER operation in [Wei and Luo, 2021]. The paper only briefly mentions in Remark 2 a subtle difference in the definition of near-stationarity compared to [Wei and Luo, 2021], but it entirely omits any discussion on the methodological and conceptual distinctions. This could lead readers to mistakenly assume that the black-box framework is an original contribution of this work. However, in reality, from the near-stationary detection rules to the integration method of the base algorithm and the format of theoretical guarantee, it bears a high degree of similarity to [Wei and Luo, 2021].

First, I believe the authors should explicitly acknowledge the inspiration and learning from [Wei and Luo, 2021] in their black-box framework. Then, they should clarify the specific methodological differences that enable the transformation of a finite-arm algorithm into an infinite-arm one. As a parallel example, [Wang, 2022], which also extends [Wei and Luo, 2021], focuses on adapting it to non-stationary bandit optimization. In that work, the author clearly states the learning and referencing of the black-box framework, as well as the specific adaptations made to suit the bandit optimization setting.

[1] Chen-Yu Wei and Haipeng Luo. Non-stationary reinforcement learning without prior knowledge: An optimal black-box approach. COLT 2021.

[2] Yining Wang. On adaptivity in nonstationary stochastic optimization with bandit feedback.

**Experimental Designs Or Analyses:**

In the experiments, it seems that only the impact of different $\beta$  of initial distributions on performance was considered, while the effect of varying non-stationarity was not validated. For instance, the regret bounds presented include two scenarios: one for piece-wise stationary settings, resulting in a $\sqrt{LT}$ bound, and another for drifting, yielding a $V^{1/3}T^{2/3}$ bound. However, the experiments appear to validate only the latter case.

**Methods And Evaluation Criteria:**

In my opinion, the methods of this work primarily extends the black-box technique from [Wei and Luo, 2021] to the infinite many-armed bandit setting, which is reasonable.

[1] Chen-Yu Wei and Haipeng Luo. Non-stationary reinforcement learning without prior knowledge: An optimal black-box approach. COLT 2021

**Other Comments Or Suggestions:**

1. The abstract does not meet the submission requirements. According to the ICML submission guidelines, abstracts must be written as a single paragraph. However, the abstract in this paper is divided into three paragraphs.

2. The paper proposes multiple algorithms and derives several regret bounds. I recommend that the authors include a table in the introduction section to summarize the algorithms and their corresponding regret bounds and compare them with the previous results. This would make it easier to understand the contributions of this work.

**Other Strengths And Weaknesses:**

The strength of this work lies in being the first to propose an optimal and parameter-free algorithm for the non-stationary infinite many-armed multi-armed bandit (MAB) setting. However, the weakness, in my opinion, is the insufficient discussion on how it distinguishes itself from prior work. For instance, both the black-box technique and the most significant arm identification method have been explored before. The paper fails to provide a clear explanation of the specific technical contributions of this work beyond these existing components.

**Questions For Authors:**

My question, as mentioned above, is specifically about understanding what the incremental part of this work is compared to existing research. For example, how exactly was the black-box technique adapted to make it suitable for the infinite-arm setting? Could the authors elaborate on the specific modifications or innovations introduced to achieve this adaptation? This would help clarify the unique technical contributions of this work.

**Relation To Broader Scientific Literature:**

Developing optimal and parameter-free algorithms for non-stationary bandits is a highly meaningful problem. Previous results have primarily focused on finite-armed multi-armed bandits (MAB) and other contextual bandit/RL scenarios, as well as bandit convex optimization (BCO) settings. This work advances the research by extending it to the infinite many-armed MAB setting.

**Theoretical Claims:**

The theoretical claims presented in this work appears to be sound, and the results align with intuition, but I have not thoroughly check the proofs.

---

> ### Author Rebuttal · Authors · 2025-04-01
>
> Thank you for the insightful review, writing suggestions, and careful discussion about the similarities with the blackbox MASTER algorithm of Wei \& Luo, 2021.
>
> >   In my opinion, the methods of this work primarily extends the black-box technique from [Wei and Luo, 2021] to the infinite many-armed bandit setting, which is reasonable.
>
> We respectfully disagree with this main point, which we believe may have lead to some confusion about the novelty of our contribution. Although our Algorithm 1 is also a blackbox, it has an entirely different algorithmic design, requirements on the base procedure, and regret analysis than MASTER. We do not agree it is a "modification or adaptation of MASTER" for the infinite-armed setting.
> We elaborate on the differences below:
> 1.  Our requirement on the base algorithm (Assumption 2) is stronger than the requirement for the base learner in MASTER (Assumption 1 of Wei & Luo, '21). Namely, we require the base algorithm to achieve instance-dependent $\log(T)/\Delta$ regret bounds in mildly non-stationary environments whereas MASTER requires its base learner to have a $\sqrt{KT}$ regret bound (see Lemma 3 of Wang, '22). This difference is quite crucial as a $\sqrt{KT}$ bound would have been insufficient for attaining the optimal regret bound in our setting (see discussion in Lines 301--308, Column 2). This difference is discussed in Remark 2 of our paper.
> 2. As you correctly observed, we rely on empirically tracking the cumulative regret of the algorithm to detect non-stationarity (which is a new idea), rather than tracking UCB indices of different base algorithms as MASTER does.
> 3. Our algorithm is conceptually different from MASTER, with **no randomized multi-scale scheduling** of different base algorithms to facilitate re-exploration of discounted arms. Instead, we run a single a base algorithm and track its cumulative regret to detect non-stationarity.
> 4. As a result, our regret analysis substantially differs from that of MASTER's, as there is no need to argue about the guarantees of the re-exploration schedule. Instead, much of the proof of Theorem 2 relies on  proving a novel high-probability regret upper bound for subsampling in mildly corrupt environments (see Lines 222-233, Column 2).
>
> Additionally, one of our main contributions is the elimination algorithm (Algorithm 2) which in fact achieves tighter regret bounds than our blackbox. Note our elimination algorithm is not a blackbox at all , and thus bears no similarity with MASTER. The only other works which study "significant arm identification" (Suk & Kpotufe, '22, '23) all again rely on randomized multi-scale scheduling of different base algorithms to target a worst-case $\sqrt{KT}$ regret bound. Our regret analysis is thus very different, with no need for analyzing random scheduling, and instead focused on novel variance-based confidence bounds for tracking empirical regret (see discussion in Sec. 5.3).
>
> We will better elaborate on these differences with the blackbox MASTER in rewrites.
>
> > In the experiments, it seems that only the impact of different $\beta$ of initial distributions on performance was considered, while the effect of varying non-stationarity was not validated. For instance, the regret bounds presented include two scenarios: one for piece-wise stationary settings, resulting in a $\sqrt{LT}$ bound, and another for drifting, yielding a $V^{1/3} T^{2/3}$ bound. However, the experiments appear to validate only the latter case.
>
> [Here](https://imgur.com/a/infinite-armed-nonstationary-bandit-experiments-Kj6VjRZ), we provide two plots of additional synthetic experiments. The second plot covers the piecewise stationary setting where we use a rotting rate of $\rho_t=1$ at $L=\sqrt{T}$ different rounds with $\beta=1$. The prior art AUCBT-ASW (Kim et al., '24) has a regret bound of $\sqrt{LT} = T^{3/4}$, yet we see from the plot that our procedures (Blackbox and Elimination) have empirically better regret owing to the small number $\tilde{L} = O(1)$ of significant shifts. We've also included an additional benchmark, a sliding-window version of SSUCB (of Bayati et al., '20) using a window size of $\sqrt{T}$, for more expansive comparison.
>
> > The abstract does not meet the submission requirements. According to the ICML submission guidelines, abstracts must be written as a single paragraph. However, the abstract in this paper is divided into three paragraphs.
>
> Thank you for pointing this out, as well as other writing suggestions. We'll fix it in revision.

---

> > ### Comment · Reviewer_T9xW · 2025-04-07
> >
> > Thanks for your detailed explanation, which helped clarify my misunderstanding. I now understand that, at the algorithmic level, the main difference from MASTER lies in the structure: MASTER maintains multiple base learners, each with different parameter settings, to estimate the unknown non-stationarity. In contrast, this work adopts an adaptive restart approach, maintaining only a single base learner and restarting it with adjusted parameters when necessary.
> >
> > Given this, I wonder if the proposed black-box method is actually more practical and computationally efficient than MASTER. I would suggest adding more comparison and discussion with MASTER in the main paper. Currently, the focus seems to be mainly on the difference in assumptions (e.g., the mild non-stationarity assumption) and the type of changes detected (as discussed in your points 1 and 2). However, this may give readers the impression that the algorithm shares the same structure as MASTER—just as I initially thought. In my view, the most important distinction is actually the one you raised in point 3, regarding the practical advantages of the algorithm.
> >
> > Additionally, I’m curious whether this type of technique could be extended to the non-stationary linear bandit setting, which also involves an infinite arm set.

---

> > > ### Author Response · Authors · 2025-04-08
> > >
> > > Thank you for your response. We will add more discussion in the rewrite highlighting the differences of our blackbox with MASTER.
> > >
> > > **MASTER vs our black-box method in terms of practicality/computational efficiency**: as said in our first response, our procedure only uses a single call to a base algorithm per epoch and thus has run-time $O(KT)$ and memory complexity $O(K)$ where $K$ is the maximum subsample size used. To contrast, MASTER (if instantiated with subsampling for our infinite-armed setting) would have a run-time of $O(KT)$ but a memory complexity of $O(KM)$ where $M$ is the maximum number of base algorithms scheduled at any one round, which is a random quantity and could take on a worst-case value of $O(\log(T))$.
> > >
> > > **Non-Stationary Linear Bandits**: a key difference between the two settings is that, in non-stationary linear bandits, the rewards of all unplayed arms change as the linear parameter $\theta_t$ changes over time. Thus, non-stationary linear bandits typically requires different techniques which tracks estimates of $\theta_t$ for non-stationarity detection. Indeed, even MASTER instantiated with OFUL implicitly does this in comparing the OFUL UCB indices of different base algorithms. We conjecture that a similar analysis as presented in this submission would go through for non-stationarity linear bandits with a fixed top reward value (e.g., $\langle a_t^*, \theta_t\rangle = 1$ for the optimal arm $a_t^*$ at time $t$) using an optimal-design based elimination algorithm. We agree this is an interesting direction for future work, and will include discussion to this extent in revisions.

---

### Official Review · Reviewer_8A6S · 2025-03-11

**Overall Recommendation:** 2

**Summary:**

This paper considers an infinite-armed bandits problem in a non-stationary setup. The means of the arms are drawn from a reservoir distribution and are chosen by an adaptive adversary in later rounds. Two algorithms are proposed with theoretical guarantees and experiments are conducted to illustrate the empirical performance.

## update after rebuttal

While the authors have resolved some of the questions, I lean to maintain my score. While the rotting cases considered in the proof of the bounds are valid (thus worst case bound is proved), I believe more general cases that consider both the rotting and rising parts of the nonstationary process are of great interest. Additionally, as mentioned by Reviewer T9xW, more comparisons with the existing method MASTER are required.

**Claims And Evidence:**

Yes, the authors provide proof outlines in sections 4 and 5 with details in the appendix.

However, in remark 1, the authors indicate Kim et al. 2022; 2024 allow for unplayed arms' rewards to change each round. As I checked these two papers, they consider **rested** rotting bandits, where only the means of the pulled arms will decrease and the means of unplayed arms remain unchanged.

**Essential References Not Discussed:**

The references look appropriate to me.

**Experimental Designs Or Analyses:**

The paper provides experimental results on some synthetic datasets, which follow the rotting bandits setup.

The experiments compare the performance of the proposed algorithms and two other algorithms in the infinite-arms setup. As there is a large amount of work on (finite-arm) non-stationary bandits, it is expected to include those algorithms as well (maybe by sampling some arms at first, followed by applying these algorithms on the sampled arms.) Also, the more general bandits setups (e.g., fluctuation, rising bandits) are expected, beyond the rotting bandits setup.

**Methods And Evaluation Criteria:**

Yes, the blackbox non-stationary algorithm transforms an algorithm for stationary bandits to adapt to the non-stationary bandits. The second algorithm tracks the significant changes in the means of the arms. These are common techniques in the non-stationary bandits literature.

**Other Comments Or Suggestions:**

None. Please refer to the above sections.

**Other Strengths And Weaknesses:**

**Strengths**
1. This paper introduces a parameter-free algorithm for the non-stationary infinite-armed bandits.
2. The blackbox framework extends the finite-armed bandit techniques to the non-stationary case.


**Weaknesses**:
1. The paper indicates the rising non-stationarity is benign and only exploit the case where the rewards are decaying (Corollary 5). Therefore, it remains unclear to me whether the bound can be further improved or not if take the rising part into consideration.

**Questions For Authors:**

Please refer to the above sections.

**Relation To Broader Scientific Literature:**

This paper considers regret minimization in the non-stationary bandits with infinite-many arms. It extends tracking the significant shifts to the infinite-armed bandits case or, extends the infinite-armed bandits to the more general non-stationary case.

**Theoretical Claims:**

I went through the proofs are the theorems. They appear reasonable to me.

The thing that confuses me is the use of $V$, $V_R$ and  $S_T$ in the regret bounds. These quantities measure the level of non-stationarity of the instances, but they also depend on the chosen arms and, consequently on the algorithm. Since different algorithms result in different values for these measurements, a direct comparison of regret bounds becomes nontrivial. For example, if an algorithm $\pi_1$ achieves a regret bound $R_{\pi_1}$ of order $O(V_{\pi_1}^{1/3})$ and another algorithm $\pi_2$ achieves $R_{\pi_2}=O(V_{\pi_2}^{1/2})$, it remains unclear whether $R_{\pi_1}<R_{\pi_2}$ or not. It is appreciated that the authors can provide more discussions on this.

In addition, for the regret lower bounds in section 3, the authors invoke the lower bounds in Kim et al. (2024), which are derived for the rotting infinite-armed bandit problem. Since this manuscript is considering the more general infinite-armed bandits problem, where the bandit instances may not be the rotting bandits case and the means of the arms are chosen by an adaptive adversary, the lower bounds for the rotting bandits cannot be directly applied here.

---

> ### Author Rebuttal · Authors · 2025-04-01
>
> Thank you for the detailed review, and astute questions.
>
> **On Comparison of Regret Bounds and $V, V_R, S_T$ Depending on Agent**:  You're correct that, in general, a direct comparison of bounds between algorithms is tricky since the adaptive adversary may vary its behavior depending on the algorithm. As a trivial example, a naive algorithm which samples a single arm and commits to it for $T$ rounds would incur linear regret $\Omega(T)$ in a stationary environment, yet could incur constant regret if an adversary helps the agent and increases the reward of the committed arm to make it optimal.
>
> We note this phenomenon is inherent to all online learning settings involving an adaptive adversary, yet there is a substantial literature of works analyzing _dynamic regret bounds_ in terms of quantities such as $V$ or $L$ (which as you note depends on the adversary and algorithm) (e.g., see [1]-[4] below).
>
> For some specific choices of adversary, a comparison of regret bounds is meaningful. For example, if the adversary is oblivious, then $V,V_R,S_T$ in our work can be upper bounded by worst-case quantities which do not depend on the agent's decisions, but on restless changes. Even for such an oblivious adversary, we note optimal and adaptive regret upper bounds were unknown before this work.
>
> Whether other meaningful choices of adversary could yield direct comparison of regret bounds is an intriguing direction for future work.
> We'll include this discussion in a revision.
>
> **On Lower Bounds in Rotting vs. General Non-Stationary Setup**: You're correct that we consider a more general setup than the rotting setup of (Kim et al., '24). Since our setup includes the rotting problem as a sub-case, the worst-case lower bounds for our setup are at least as large as the worst-case lower bounds for the rotting sub-case. Thus, the lower bounds of Kim et al., '24 hold for our setting as well.
>
> **On Broader Experimental Comparison with Other Algorithms and Setups:** We emphasize our main contribution is theoretical, rather than to propose a practical algorithm. The algorithms in our paper are of a theoretical nature that serves to resolve the main question of attaining the first optimal and adaptive regret bound for non-stationary infinite-armed bandits. The synthetic experimental results of Section 6 are mainly to support the message that our algorithm does indeed attain improved regret over the previous state-of-art. We agree with the reviewer that there's much further work to be done in designing more practical procedures, and admit the theoretical state-of-the-art is far from this.
>
> We'd also like to note that the strategy of "sampling a fixed set of arms and then running a finite-armed non-stationary bandit algorithm'' would not have the correct theoretical regret upper bounds as the optimal sampling rate depends on the unknown non-stationarity and, furthermore, only worst-case rates of the form $\sqrt{LKT}$ for $K$ sampled arms are known for such procedures (e.g., Wei & Luo, '21), which would be inadequate for attaining the optimal regret rate (see discussion in Lines 301--308, Column 2). The difficulty of using this approach is also further discussed in Lines 112--140 (Column 2) of the paper.
>
> >   The paper indicates the rising non-stationarity is benign and only exploit the case where the rewards are decaying (Corollary 5). Therefore, it remains unclear to me whether the bound can be further improved or not if take the rising part into consideration.
>
> We'd like to emphasize that we _allow_ for both rising and rotting non-stationarity. You are correct, however, that the bound of Corollary 5 is worst-case and only takes into account the challenge of rotting changes. It is an interesting future direction to study whether rising can improve the rate. Note, even in finite-armed bandits, improved regret bounds beyond the worst-case $\sqrt{LT} \land V^{1/3} T^{2/3}$ rates are still unknown in general.
>
> > However, in remark 1, the authors indicate Kim et al. 2022; 2024 allow for unplayed arms' rewards to change each round. As I checked these two papers, they consider rested rotting bandits, where only the means of the pulled arms will decrease and the means of unplayed arms remain unchanged.
>
> Thanks for catching this! You're correct that Kim et al. ('22, '24) in fact study the same rested setting as ours, and we'll revise this remark.
>
> [1] Peter Auer, Nicolo Cesa-Bianchi, Yoav Freund, and Robert E Schapire. The nonstochastic multiarmed bandit problem. SIAM
> journal on computing, 2002.
>
> [2] Ali Jadbabaie, Alexander Rakhlin, Shahin Shahrampour, and Karthik Sridharan. Online Optimization : Competing with Dynamic
> Comparators. AISTATS 2015.
>
> [3] Tianbao Yang, Lijun Zhang, Rong Jin, and Jinfeng Yi. Tracking slowly moving clairvoyant: Optimal dynamic regret of online
> learning with true and noisy gradient. ICML, 2016.
>
> [4] Peng Zhao, Yu-Jie Zhang, Lijun Zhang, and Zhi-Hua Zhou. Dynamic regret of convex and smooth functions. NIPS ’20

---

### Official Review · Reviewer_onxF · 2025-03-11

**Overall Recommendation:** 3

**Summary:**

This manuscript deals with the problem of regret minimization in an infinite-arm bandit model with a reservoir distribution where shifts can occur. As such, this work is at the crossroad of infinite bandit and non-stationary bandits. Recently, Kim et al. (2024) have characterized the minimax cumulative regret as a function of the reservoir distribution of non-stationarity indices. However, their procedure required the knowledge of these non-stationarity indices (namely V_R and T_R). In this work, the authors introduce two rate-optimal procedures which only require the knowledge of the reservoir distribution. The procedures are based on a combination of three ideas: (i) using, as classical in this field, subsampling techniques to work with a finite arm problem (ii) tracking the empirical regrets/arm reward (which is estimable as the optimal reward is known) and (iii) suitably reinitializing the sampling process when the empirical regret is unusually large or all arms are unusually large.


## update after rebuttal
In their rebuttal, the authors have well addressed my questions. Nevertheless, this does not change significantly my opinion on this work.

**Claims And Evidence:**

The manuscript is clearly written and all the proofs seem to be correct.

**Essential References Not Discussed:**

Not really, but the authors should emphasize that the "subsampling idea" for infinite-arms bandit is much older than Bayati et al[2020], see the earlier work of Berry. It is in fact standard in the infinite arm field.

**Experimental Designs Or Analyses:**

The small-scale numerical experiments illustrate well the theoretical findings.

**Methods And Evaluation Criteria:**

The proposed procedures are minimax adaptive with respective to the cumulative regret which is the standard way of  evaluating the performances for a bandit problem.

**Other Comments Or Suggestions:**

As such, hypotheses are lacking for Theorem 4 and Corollary 5. If the authors do not want to rely on Assumption 1, they should at least introduce a dedicated assumption.

In Section 1.2, the authors emphasize that this is the first work to develop dynamic regret bounds of the $V^{1/3}T^{2/3}\wedge \sqrt{LT}$, while such result are unknown in the finite-arm setting. I would simply like to point out that the infinite-arm setting has distinct feature which can make the problem easier than finite arm: the optimal reward is known and the distribution of the mean rewards is also known.

**Other Strengths And Weaknesses:**

S1) On the positive side,  this is the first work which is adaptive to the changes in the infinite arm regime. Besides, the new algorithms are quite simple and natural.

W1) The counterpart of this strength is that the key ideas for the algorithms are not so original, although such subsampling techniques appear to be new in for non-stationary infinite-arms bandits.

W2) I feel that the topic is perhaps not that important so that achieving adaptivity (while the optimal rate were already known) makes a big impact.

**Questions For Authors:**

In Theorem 2, the authors provide a cumulative regret bound for Algorithm 2 which rely that use, as black-box, a bandit algorithm for $\alpha$-mildly correct bandits. How does the bound in Theorem 2 depends on $\alpha$? Alternatively, should the black-box satisfy the bound of Assumption 2 for any $\alpha$?

**Relation To Broader Scientific Literature:**

The main contribution of this work is adaptivity to non-stationarity in infinite-arm bandits. This improves over the recent work of Kim et al. [2024]. For that purpose, the authors mainly adapt a sucessive elimination algorithm from (Even-Dar et al, 2006). As an aside, the proofs built upon previous arguments of Suk for non-stationary multi-armed bandits.

**Theoretical Claims:**

I checked all the proof sketches and all the general arguments in the proof but I did not check all computational details. There does not seem to be any typo.

---

> ### Author Rebuttal · Authors · 2025-04-01
>
> Thank you for the detailed review and positive comments.
>
> > the authors should emphasize that the "subsampling idea" for infinite-arms bandit is much older than Bayati et al[2020], see the earlier work of Berry. It is in fact standard in the infinite arm field.
>
> This is correct - subsampling was introduced earlier for this problem, and we'll adjust the writing to reflect this.
>
> > W1) The counterpart of this strength is that the key ideas for the algorithms are not so original, although such subsampling techniques appear to be new in for non-stationary infinite-arms bandits.
>
> Indeed, you're correct that subsampling was not used in the prior works on non-stationary infinite-armed bandits (Kim et al., '22, '24). We'd like to highlight two additional technical innovations required for our result:
> 1. Tracking cumulative regret using variance-based confidence bounds (this was crucial for attaining the optimal regret bound for $\beta < 1$ which was not achieved in previous works _even with parameter knowledge_). Note the previous works on adaptive non-stationary finite-armed bandits (e.g., Wei & Luo '21; Suk & Kpotufe '22) don't require such a refined analysis as their worst-case regret rates scale with the number of arms.
> 2. Doing a high-probability per-arm regret analysis for subsampling (i.e., Lines 222--233, Column 2) which is novel and departs from the previous regret analysis for subsampling (e.g., Bayati et al., '20).
>
> > W2) I feel that the topic is perhaps not that important so that achieving adaptivity (while the optimal rate were already known) makes a big impact.
>
> As we discussed in the previous answer, in fact the optimal regret upper bound was not known for $\beta < 1$ as there was a gap between upper and lower bounds even with knowledge of non-stationarity. Our work closes this gap.
>
> > As such, hypotheses are lacking for Theorem 4 and Corollary 5. If the authors do not want to rely on Assumption 1, they should at least introduce a dedicated assumption.
>
> Theorem 4 and Corollary 5 rely on Assumption 1, but without the upper bound on masses involving $\kappa_2$.
> We'll make this more clear in a revision.
>
> > In Section 1.2, the authors emphasize that this is the first work to develop dynamic regret bounds of the $V^{1/3} T^{2/3} \land \sqrt{LT}$, while such result are unknown in the finite-arm setting. I would simply like to point out that the infinite-arm setting has distinct feature which can make the problem easier than finite arm: the optimal reward is known and the distribution of the mean rewards is also known.
>
> This is an apt point that the two settings are not directly comparable. We contend that the infinite-armed setting is not necessarily "easier" than the finite-armed counterpart. For instance, the optimal regret bounds in our setting must be free of the number of arms and so a crude $\sqrt{KT}$ bound cannot be plugged in for subsampling, instead necessitating tighter instance-dependent or variance-based bounds (as can be found in our work). In comparison, the $\sqrt{LKT}$ regret bound in $K$-armed non-stationary bandits does not require such refined analyses.
>
> > In Theorem 2, the authors provide a cumulative regret bound for Algorithm 2 which rely that use, as black-box, a bandit algorithm for $\alpha$-mildly correct bandits. How does the bound in Theorem 2 depends on $\alpha$? Alternatively, should the black-box satisfy the bound of Assumption 2 for any $\alpha$?
>
> As our goal is to attain the optimal regret bound of $\sqrt{LT} \land V^{1/3} T^{2/3}$, we in fact only require Assumption 2 to hold in each episode $[t_{\ell}, t_{\ell+1})$ for $\alpha \approx (t_{\ell+1} - t_{\ell})^{-\frac{1}{\beta+1}}$.
> In terms of $\alpha$, we then show a regret bound of $\sum_{\ell=1}^{\hat{L}} (t_{\ell+1} - t_{\ell}) \cdot \alpha$ where $\hat{L}$ is the total number of episodes. Thus, for Theorem 2 to hold, it is **not** required for Assumption 2 to be true for any $\alpha$. Nevertheless, we show in Appendix C that classical algorithms such as UCB do in fact satisfy Assumption 2 for any $\alpha$.

---

### Decision · Program_Chairs · 2025-05-01

**Decision:**

Accept (poster)

**Comment:**

This submission, pertaining to infinite-armed reservoir bandits with non-stationary distribution shifts, has received three reviews, with final scores in the borderline-to-positive region.

The referees initially raised concerns related to attribution of certain algorithmic ideas to prior work, connections to the rotting bandits literature, alternative algorithmic approaches based on subsampling, comparisons with the previously known blackbox-MASTER algorithm for non-stationary finite-armed bandits. The authors have provided detailed and adequate responses clarifying the connection to previous work, the inadequacy of pure-subsampling based approaches, and contrast with the MASTER algorithm.

In particular, two reviewers (T9xW and 8A6S) raised concerns about the comparison with MASTER, but the authors have later clarified satisfactory and in detail that their algorithm is different from MASTER on several counts, including a simpler structure avoiding multi-scale scheduling of subroutines.

While some concerns, including the lack of more specific structural analyses in terms of the rising / rotting profile of the non-stationarity, persist in the final stage, I am inclined to believe that this paper introduces new and useful ideas, both from an algorithmic and analytical perspective, to the domain of infinite-armed non-stationary bandits. In view of this, I recommend acceptance of the paper.